# MUR: Momentum Uncertainty Guided Reasoning For Large Language Models

## Abstract

Large Language Models (LLMs) have achieved impressive performance on reasoning-intensive tasks, yet optimizing their reasoning efficiency remains an open challenge. While Test-Time Scaling (TTS) improves reasoning quality, it often leads to overthinking—wasting tokens on redundant computations. This work investigates *how to efficiently and adaptively guide LLM TTS **without additional training***. Inspired by the concept of momentum in physics, we propose Momentum Uncertainty-guided Reasoning (*MUR*), which dynamically allocates thinking budgets to critical reasoning steps by tracking and aggregating step-wise uncertainty over time. To support flexible inference-time control, we introduce $\gamma$-control, a simple mechanism that tunes the reasoning budget via a single hyperparameter. We provide theoretical intuition to support the superiority of *MUR* as a low-pass filter. *MUR* is comprehensively evaluated against various TTS methods across four challenging benchmarks (MATH-500, AIME24, AIME25, and GPQA-diamond) using different sizes of recent Qwen3 models (1.7B, 4B, and 8B). Results demonstrate that *MUR* reduces computation by over 45% on average while improving accuracy by 0.33–3.46%.

## 1 Introduction

Large Language Models (LLMs) (Brown et al., 2020; Grattafiori et al., 2024) demonstrate remarkable performance in reasoning-intensive scenarios, including logic, mathematics, and game-playing tasks. A critical advancement in optimizing their reasoning quality is *Test-Time Scaling* (TTS). Existing methods either incentivize long thinking patterns through reinforcement learning with verifiable rewards (RLVR) (Ye et al., 2025; Jaech et al., 2024; Guo et al., 2025), or employ stepwise optimization via parallel sampling (Yao et al., 2023; Lightman et al., 2023; Wang et al., 2024b; Ma et al., 2024; Xu et al., 2025) and sequential critique (Lan et al., 2024; Li et al., 2025).

While effective, the issue of *overthinking* (Chen et al., 2024b; Sui et al., 2025) is widely observed that degrades the inference efficiency. As shown in Figure 1, the performance can

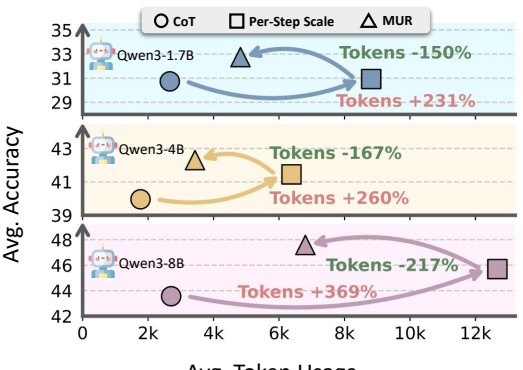

Figure 1: Comparisons between average accuracy and token usage. *Per-Step Scale* refers to test-time scaling methods that optimize every step without compute-saving mechanisms. *MUR* is a computationally efficient approach that selectively scales only key steps. The percentage in this figure is calculated based on CoT budget without TTS.

even be slightly improved, despite >45% reduction in thinking tokens against Per-Step Scale. This demonstrates that there is significant room for improvement in making long thinking concise.

Intuitively, LLMs should spend more token budgets on complex steps to deliberately enhance output quality, while generating simple steps directly to avoid overthinking. However, it still remains challenging to identify key steps and dynamically allocate computes. Recent works (Xia et al., 2025a; Jiang et al., 2025; Yang et al., 2025d; Yu et al., 2025; Yang et al., 2025c) explore training methods to adaptively allocate token usage on different steps, which introduce additional training costs and lack

generalization. Off-the-shelf training-free methods (Kim et al., 2025; Xu et al., 2025; Wang et al., 2025) scale thinking tokens in a fixed manner, failing to adapt to problem complexity or on-going reasoning process.

Therefore, the pursuit of efficiently and adaptively guiding LLM test-time scaling without extra-training is both intriguing and understudied. To answer this question, we are the first to model LLM reasoning with the concept of momentum. In physics, momentum accumulates historical information over time and resists sudden changes. Based on this and the successful application of Gradient Descent with Momentum (Qian, 1999), we propose Momentum Uncertainty guided Reasoning (*MUR*), a novel approach that dynamically evaluates the overall uncertainty of a reasoning path by aggregating historical step-level uncertainties, mirroring the smooth and consistent evolution observed in physical dynamics. Without requiring any training, *MUR* selectively allocates computation only to critical steps during inference. Based on the approach, we introduce the concept of $\gamma$-control, where we can flexibly control the thinking budget and the performance, with only one hyperparameter $\gamma$. Further, this work proves that *MUR* is theoretically grounded in terms of discounted credit assignment and stability while maintaining compatibility with existing TTS methods. Extensive experiments across four challenging benchmarks and three backbone model sizes demonstrate that *MUR* reduces the thinking budget by over 45% on average while even improving accuracy by 0.33–3.46%.

The key contributions include:

**(1) Adaptive Scaling Technique**. We propose the novel concept of momentum uncertainty and offer a training-free solution *MUR* to dynamically allocate thinking budgets to key reasoning steps guided by momentum uncertainty, which is compatible with various TTS methods.

**(2) Efficiency and Performance Gains**: *MUR* reduces the thinking costs by 45% even with obvious performance gains, across a wide range of benchmarks and model sizes. The proposed $\gamma$-control offers flexible solution to balance performance and efficiency.

**(3) Theoretical Support**: *MUR* is theoretically grounded in terms of discounted credit assignment, stability, and convergence, which support its practical superiority.

## 2 RELATED WORK

### 2.1 TEST-TIME SCALING

Test-Time scaling (TTS) methods allocate additional token usage during inference, revealing a scaling law (Brown et al., 2024; Wu et al., 2024) that more computes lead to better performance. Training-based methods elicit long thinking patterns through reinforcement learning with verifiable rewards (RLVR) (Ye et al., 2025; Jaech et al., 2024; Guo et al., 2025). Training-free methods can be categorized into parallel scaling and sequential scaling. Parallel scaling (Yao et al., 2023; Ma et al., 2024; Xu et al., 2025) samples several answers for the same input, followed by selecting the best one. Sequential scaling (Lan et al., 2024; Li et al., 2025) utilizes feedback from self-evaluation or external models to optimize current answer. Although these researches show remarkable achievements, they allocate unnecessary computes for simple steps. Our work *MUR*, as an orthogonal method to these researches, optimizing these methods by guiding them to scale only key steps, reducing unnecessary computes largely.

### 2.2 OVERTHINKING

Although LLMs demonstrate significant performance gains through test-time scaling methods, they are likely to introduce computational overhead and reasoning latency (Chen et al., 2024b; Sui et al., 2025). One line of mitigating overthinking is to shorten reasoning length through post-training (Xia et al., 2025a; Jiang et al., 2025; Yang et al., 2025d; Yu et al., 2025; Yang et al., 2025c), which introduces training overhead and limits their generalization. Another line is training-free methods (Kim et al., 2025; Xu et al., 2025; Wang et al., 2025), reducing token usage in a fixed manner, which lacks adaptation to on-going reasoning process. Our work *MUR*, without training, adaptively saves unnecessary computes during the whole reasoning process.

### 2.3 UNCERTAINTY ESTIMATION

The reasoning path of LLM often contains reliability issues, like hallucinations or biased responses (Xia et al., 2025b). One line of uncertainty estimation is scaling more computes, including verbalizing methods (Tian et al., 2023; Tanneru et al., 2024), consistency-based methods (Hou et al., 2024; Chen & Mueller, 2024; Gao et al., 2024), and semantic clustering methods (Kuhn et al., 2023; Farquhar et al., 2024; Nikitin et al., 2024). Another line of uncertainty estimation is utilizing the internal information during decoding (Ahdritz et al., 2024; Chen et al., 2024a; Sriramanan et al., 2024), which estimates the uncertainty of generated path through aggregating token-level probabilities, lacking the adaptation to different reasoning steps. Our method *MUR*, assigns more attention to recent steps, while reducing the impact of early steps.

## 3 METHOD

In this section, we first formulate the stepwise test-time scaling, adaptive scaling and step-level uncertainty (Sec. 3.1). Then we formally propose momentum uncertainty, followed by theoretical proof of its superiority (Sec. 3.2). Based on the momentum uncertainty, we introduce $\gamma$-control mechanism to flexibly scale inference-time scaling (Sec. 3.3). The overview of *MUR* is presented in Figure 2.

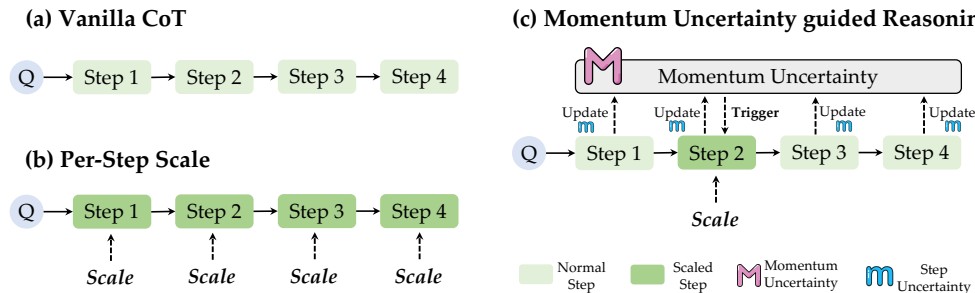

Figure 2: Comparison of reasoning methods. (a) *Vanilla CoT*: Standard stepwise reasoning without test-time scaling. (b) *Per-Step Scale*: scales computes per reasoning step. (c) *MUR*: Adaptive test-time scaling framework (ours).

### 3.1 PRELIMINARY

**Stepwise test-time scaling** LLM reasoning can be formulated as auto-regressively generating step $a_t$ at each timestamp $t$, based on the inputs and previous steps:

$$a_t \sim p_\theta(\cdot|x, \mathbf{a}_{<t}), \tag{1}$$

where $x$ is the concatenation of input question and instruction. $\mathbf{a}_{<t}$ represents previous steps . $\theta$ denotes the parameters of pre-trained LLM, and $p_\theta$ is the probability distribution.

To optimize the quality of the reasoning path, current methods apply test-time scaling at each step, which can be formulated as follows:

$$\hat{a}_t \sim Q(\cdot|x, \mathbf{a}_{<t}), \tag{2}$$

where $\hat{a}_t$ is the optimized step. $Q$ denotes the specific test-time scaling method, such as *Best-of-N* (Brown et al., 2024).

**Adaptive Scaling** Conventional test-time scaling methods typically apply optimization at every decoding step, leading to excessive token usage and computational overhead. However, not all steps require such enhancement, and current research on adaptive compute allocation remains limited, often overlooking this inefficiency. We therefore pose the central question: ***When should compute be scaled during inference?*** To address this, we model this research question with a binary detector

$D$ that selectively activates test-time scaling based on contextual reasoning dynamics:

$$\hat{a}_t = \begin{cases} Q(\cdot|x, \mathbf{a}_{<t}) & , D(t) = \text{True} \\ a_t & , D(t) = \text{False} \end{cases}.$$ (3)

Here, $D$ determines whether to invoke a test-time scaling method at each step based on historical information. Our work focuses **exclusively** on designing the detector $D$ to assess the reasoning trajectory and adaptively decide whether to allocate additional compute to the current step $a_t$.

**Step-level Uncertainty** Uncertainty estimation quantifies an LLM's confidence in its output, where higher uncertainty implies lower confidence. For step $a_t$ consisting of $N$ tokens, we compute the step-level uncertainty based on token-wise probabilities. Specifically, we define the average negative log-likelihood of the tokens as:

$$m_t = \frac{1}{N} \sum_{j=1}^{N} -\log p_\theta(a_t^{(j)}|x, \mathbf{a}_{<t}, a_t^{(<j)}),$$ (4)

where $m_t$ is the uncertainty of step $t$. $a_t^{(j)}$ is $j$-th token of step $a_t$. And $a_t^{(<j)}$ denotes the prefix token sequence $a_t^{(1)}, a_t^{(2)}, ..., a_t^{(j-1)}$.

## 3.2 MOMENTUM UNCERTAINTY

LLM can maintain an uncertainty estimation $M$ for the reasoning process, reflecting the global assessment of both input $x$ and generated steps $\mathbf{a}_{\leq t}$. Ideally, this uncertainty should evolve smoothly, adapting to new steps as they are generated, while preserving a calibrated estimate of earlier steps. Inspired by the concept of momentum in physics, which retains and updates an object's motion by accumulating past forces while resisting abrupt changes. We propose momentum uncertainty, a recursive formulation of $M$ that dynamically tracks overall uncertainty during reasoning:

$$M_t = \alpha M_{t-1} + (1 - \alpha)m_t,$$ (5)

where $M_t$ is the momentum uncertainty at timestamp $t$, with initial value $M_0 = 0$. And $\alpha \in (0, 1)$ is a hyper-parameter controlling the momentum changing.

With a recursive definition, momentum uncertainty aggregates all generated step-level uncertainties to represent the overall estimation of the reasoning process. Further, we introduce the excellent property of *momentum uncertainty* with theoretical and experimental analysis.

**Proposition 1:** *Momentum uncertainty is an exponentially weighted sum of step-level uncertainties, emphasizing recent steps and fading earlier ones.*

*Proof.* We provide a detailed derivation in Appendix A.1. It transforms Equation 5 into the exponential weighting of step-level uncertainties as follows:

$$M_t = (1 - \alpha) \sum_{i=1}^{t} \alpha^{t-i} m_i.$$ (6)

Through Equation 6, $M_t$ assigns different weights $\alpha^{t-i}$ to historical step-level uncertainty $m_i$, emphasizing recent uncertainties while smoothing early fluctuations, balancing the attention among different steps. This aligns with the intuition that recent steps can better represent the reasoning uncertainty, so that momentum uncertainty can well track the evolving of uncertainty change.

Notably, We focus solely on the internal uncertainty signals of the model, disregarding the specific logical information of the output content. This is because the uncertainty signal inherently reflects the accuracy of the model's reasoning (Xu et al., 2025; Yang et al., 2025b). □

**Proposition 2:** *Acting as a low-pass filter, momentum uncertainty $M_t$ attenuates high-frequency components while preserving low-frequency signals, leading to more stable estimates.*

*Proof.* LLM decoding contains unavoidable noise (Wang et al., 2024a; Zhou et al., 2024), introducing variance to uncertainty estimation. Assume each step-level uncertainty $m_t$ contains two parts:

$$m_t = \mu_t + \epsilon_t, \tag{7}$$

where $\mu_t$ is the pure step-level uncertainty, and $\epsilon_t$ is a noise originating from training or randomly sampling, etc.

Leveraging the frequency-domain framework of Li et al. (2024) and the convergence theory of Liu et al. (2020), we can treat the momentum uncertainty as a low-pass filter as follows:

$$H(\omega) = \frac{1 - \alpha}{1 - \alpha e^{-j\omega}}, \tag{8}$$

where $\omega \in [0, \pi]$ denotes the normalized signal angular frequency. And the derivative of the magnitude response is as follows:

$$\frac{d|H(\omega)|}{d\omega} = -\frac{(1 - \alpha)\alpha \sin \omega}{(1 - 2\alpha \cos \omega + \alpha^2)^{3/2}} < 0, \tag{9}$$

so the magnitude response $|H(\omega)|$ decreases monotonically from 1 to $\frac{1-\alpha}{1+\alpha}$ as $\omega$ increases from 0 to $\pi$, demonstrating low-pass filter behavior, which can effectively attenuate high-frequency components $\epsilon_t$. The high-frequency signal contains noise and sudden fluctuation of reasoning uncertainty, both of which will be filtered to smooth the estimation process of reasoning uncertainty $\mu_t$. Detailed proof is attached in Appendix A.2.

$\square$

While the auto-regressive nature of LLMs leads to a theoretical expectation of temporal correlation in the noise signal, our empirical findings justify the validity of independent modeling. We analyze the autocorrelation function (ACF) of the noise signal using real data sampled from several LLMs, and results demonstrate that, we have confidence over 95% to consider real noise signal $\epsilon_t$ is not temporally correlated. Based on this critical finding, we provide further theoretical intuition and experimental analysis that momentum uncertainty is superior to naive average uncertainty method (Ren et al., 2022; Manakul et al., 2023; Dobriban et al., 2024). More details can be found in A.3. Moreover, experimental comparison is in Sec. 4.2.

### 3.3 SCALABLE THINKING WITH $\gamma$-CONTROL

Since momentum uncertainty captures the overall confidence in the reasoning trajectory, we propose a $\gamma$-control mechanism to identify whether the current step is incompatible with prior reasoning. This mechanism balances reasoning performance against computational cost.

**Scale High-uncertainty Steps** At each step, the step-level uncertainty $m_t$ reflects the model's confidence in the current generation $a_t$, while $M_{t-1}$ aggregates uncertainty over previous steps. If $m_t > M_{t-1}$, the current step is more uncertain than the reasoning history, suggesting it may be erroneous. To address this, we introduce a checking mechanism that selectively scales uncertain steps.

To tolerate minor fluctuations while flagging significant deviations, we apply a $\gamma$-*control* threshold. Specifically, we define a detector $D$ in Equation 3 as:

$$\hat{a}_t = \begin{cases} Q(\cdot|x, \mathbf{a}_{<t}) & , \exp(m_t) > \exp(M_{t-1})/\gamma \\ a_t & , \text{others} \end{cases}, \tag{10}$$

where $\gamma$ is the controllable scaling rate, ranging from (0,1) in practice. The scaling factor $\frac{1}{\gamma}$ effectively raises the detection boundary, allowing slight uncertainty increases while catching large deviations. Smaller $\gamma$ values result in fewer steps being scaled, enabling flexible control over the computational budget. More details can be found in Appendix.

The inequality in Equation 10 flags when a step diverges significantly from the previous reasoning, a corrective test-time scaling is triggered to improve output quality. A theoretical analysis of $\gamma$-control is provided in Appendix A.4 and empirical results of $\gamma$-control is presented in Sec. 5.1.

**Orthogonal to Test-Time Scaling Methods**  Our momentum uncertainty-based detector $D$ is orthogonal and complementary to current test-time scaling methods, such as *best-of-N* and thinking model. It identifies uncertain steps and selectively triggers compute-intensive optimization, maintaining or even improving overall performance while reducing redundancy.

## 4  EXPERIMENTS

### 4.1  EXPERIMENTAL SETUP

**Benchmarks**  We evaluate our proposed method *MUR* on three widely adopted math reasoning benchmarks MATH-500 (Hendrycks et al., 2021), AIME24, and AIME25. In addition, we include GPQA-diamond (Rein et al., 2024) to validate the generalization to the science domain.

**Metrics**  We adopt pass@1 rate as our **Acc.** metric. We also report the average token usage of backbone model as **#Token** for each solution, providing an aspect of efficiency evaluation. For AIME24 and AIME25, to reduce the infection of randomness, we sample 16 times for each query and report the average accuracy and token usage.

**Test-Time Scaling Settings**  We adopt four test-time scaling methods as the basic setting. 1) *Guided Search*. It can be viewed as step-level *Best-of-N* (Brown et al., 2024), where $N$ candidate steps are sampled in parallel at each timestep, and the optimal one is selected. We adopt GenPRM (Zhao et al., 2025) as an external reward model for candidate selection. 2) *LLM As a Critic*. The LLM receives feedback after generating each step and iteratively refines its output based on the critique (Lan et al., 2024; Li et al., 2025). We also adopt GenPRM for stepwise feedback generation. 3) $\phi$-*Decoding* (Xu et al., 2025). It does not require external models but selects the best step from several candidates using the foresight sampling strategy. 4) *Thinking Mode* (Yang et al., 2025a) Models with thinking mode generates longer reasoning path, introducing deliberate optimization to each step.

**Baselines**  We adopt four baselines. 1) *CoT* (Wei et al., 2022). Standard stepwise reasoning without scaling. 2) *Per-Step Scale*. Test-time scaling methods that scale the computation for each step. 3) *Avg. uncertainty*. Average the uncertainty across all generated steps (Ren et al., 2022; Manakul et al., 2023; Dobriban et al., 2024) to represent the overall uncertainty of the reasoning process, then scale steps with uncertainty higher than this average. 4) *SMART*. Following the original work by Kim et al. (2025), the backbone model generates reasoning steps autonomously. If the token-level confidence (TLC) falls below a predefined threshold, we apply TTS methods.

**Implementation Details**  We conduct all experiments on different models from Qwen3-series (Yang et al., 2025a), including Qwen3-1.7B, Qwen3-4B, and Qwen3-8B. The hyperparameter $\alpha$ and $\gamma$ are both set to 0.9 as default if no additional explanation is provided. For more implementation details, please refer to Appendix B.

### 4.2  MAIN RESULTS

Table 1 and Table 2 report four widely adopted reasoning benchmarks across 3 sizes of models.

***MUR* consistently outperforms strong baselines.**  The main results demonstrate the superior token saving capacity of *MUR* in most scenarios, and consistently improves the accuracy against Per-Step Scale methods (from 0.33% to 3.46%). This benefits from reducing overthinking on simple steps, while keeping optimization for difficult steps.

*MUR* outperforms average uncertainty and SMART on both token usage and accuracy (1.66%, 1.62% for average, respectively). Although the two baselines generate fewer tokens than *MUR* in

Table 1: Main results. The best results are highlighted in bold. **Acc.** denotes pass@1 rate and **#Tokens** denotes the **backbone model's** average token usage for each query, more details concerning external model token usage is in Appendix C.1. We also report the delta compared to *Per-Step Scale* baseline, including the accuracy difference and the percentage of saved tokens. Red indicates worse performance, while green indicates better performance against Per-Step Scale. Here, ↑ denotes that higher values are better, whereas ↓ means lower values are preferable.

| | MATH-500 | | AIME24 | | AIME25 | | GPQA-diamond | | Avg. | | | |
|---|---|---|---|---|---|---|---|---|---|---|---|---|
| | Acc.↑ | #Tokens↓ | Acc.↑ | #Tokens↓ | Acc.↑ | #Tokens↓ | Acc.↑ | #Tokens↓ | Acc.↑ | Δ↑ | #Tokens↓ | Δ↓ |
| *Qwen3-1.7B* | | | | | | | | | | | | |
| **Vanilla CoT** | 69.20 | 1,047 | 17.92 | 4,243 | 9.58 | 4,273 | 26.26 | 1,086 | 30.74 | - | 2,662 | - |
| **Guided search** | | | | | | | | | | | | |
| + Per-Step Scale | 70.80 | 3,460 | 17.92 | 17,463 | 10.42 | 16,680 | 27.27 | 6,739 | 31.60 | - | 11,086 | - |
| + Avg uncertainty | 70.20 | 2,398 | 18.33 | 7,850 | 9.58 | 8,883 | 25.76 | 3,404 | 30.97 | (-0.63) | 5,634 | (-49.18%) |
| + SMART | 70.80 | 3.128 | 17.50 | 8,955 | 8.96 | 10,091 | 24.74 | 3,825 | 30.50 | (-1.10) | 6,500 | (-41.37%) |
| + *MUR* (ours) | 71.20 | 1,321 | 18.33 | 4,712 | 10.63 | 5,179 | 32.83 | 2,005 | 33.25 | (+1.65) | 3,304 | (-70.19%) |
| **LLM as a critic** | | | | | | | | | | | | |
| + Per-Step Scale | 70.20 | 1,098 | 16.04 | **3,362** | 10.00 | **3,160** | 28.28 | **892** | 31.13 | - | 2,128 | - |
| + Avg uncertainty | 68.60 | 1,019 | 17.92 | 4,176 | 9.17 | 3,174 | 26.77 | 1,417 | 30.62 | (-0.51) | **2,447** | (+14.97%) |
| + SMART | 70.40 | **878** | 18.96 | 3,976 | 8.96 | 3,600 | 28.28 | 1,446 | 31.65 | (+0.52) | 2,475 | (+16.31%) |
| + *MUR* (ours) | 71.20 | 902 | **19.38** | 3,892 | **10.21** | 4,011 | 32.32 | 1,693 | 33.28 | (+2.15) | 2,625 | (+26.25%) |
| **φ-Decoding** | | | | | | | | | | | | |
| + Per-Step Scale | 68.00 | 5,501 | 17.50 | 19,612 | 8.96 | 18,550 | 25.76 | 9,261 | 30.06 | - | 13,231 | - |
| + Avg uncertainty | 69.00 | 2,844 | 19.17 | 13,743 | 8.33 | **15,785** | 25.25 | 2,431 | 30.44 | (+0.38) | 8,701 | (-34.24%) |
| + SMART | **70.20** | 3,848 | **21.04** | 19,437 | 8.13 | 24,113 | 23.23 | 3,338 | 30.65 | (+0.59) | 12,684 | (-4.13%) |
| + *MUR* (ours) | 69.80 | **2,520** | 20.21 | **13,711** | 9.58 | 16,088 | 27.27 | 1,827 | 31.72 | (+1.66) | 8,537 | (-35.48%) |
| *Qwen3-4B* | | | | | | | | | | | | |
| **Vanilla CoT** | 79.40 | 772 | 24.08 | 3,111 | 16.46 | 2,577 | 39.90 | 612 | 39.96 | - | 1,768 | - |
| **Guided search** | | | | | | | | | | | | |
| + Per-Step Scale | 79.80 | 3,048 | 29.38 | 13,761 | **19.17** | 10,663 | 42.42 | 3,517 | 42.69 | - | 7,747 | - |
| + Avg uncertainty | 79.80 | 1,911 | 28.33 | 7,012 | 18.54 | 7,719 | 39.90 | 1,354 | 41.64 | (-1.05) | 4,499 | (-41.93%) |
| + SMART | **81.60** | 2,476 | 24.58 | 8,515 | 15.42 | 9,375 | **43.43** | 2,116 | 41.26 | (-1.43) | 5,621 | (-27.45%) |
| + *MUR* (ours) | 81.40 | **824** | 29.58 | 4,265 | **19.17** | 7,162 | 41.92 | 929 | 43.02 | (+0.33) | 3,295 | (-57.47%) |
| **LLM as a critic** | | | | | | | | | | | | |
| + Per-Step Scale | 80.80 | 777 | 25.21 | 3,334 | 17.92 | 3,260 | 40.91 | 737 | 41.21 | - | 2,027 | - |
| + Avg uncertainty | 81.40 | **741** | 25.63 | 3,217 | 20.00 | 3,120 | 39.90 | 804 | 41.73 | (+0.52) | **1,971** | (-2.79%) |
| + SMART | 80.60 | 813 | **26.04** | 3,203 | 17.50 | 3,201 | **43.43** | 724 | 41.89 | (+0.68) | 1,985 | (-2.06%) |
| + *MUR* (ours) | **81.60** | 745 | **26.04** | 3,309 | **20.21** | **3,113** | 40.91 | 699 | 42.19 | (+0.98) | 1,967 | (-2.98%) |
| **φ-Decoding** | | | | | | | | | | | | |
| + Per-Step Scale | 76.80 | 4,690 | 27.08 | 14,394 | 16.46 | 14,109 | 41.41 | 4,263 | 40.44 | - | 9,364 | - |
| + Avg uncertainty | **80.60** | 1,866 | 26.67 | 14,361 | **18.54** | 14,836 | 39.90 | 1,511 | 41.43 | (+0.99) | 8,144 | (-13.03%) |
| + SMART | 79.40 | 2,776 | 26.25 | 19,327 | 17.71 | 22,807 | 40.40 | 2,195 | 40.94 | (+0.50) | 11,776 | (+25.76%) |
| + *MUR* (ours) | 79.60 | **1,796** | 27.29 | 8,563 | 18.13 | 8,845 | 41.92 | 944 | 41.74 | (+1.30) | 5,037 | (-46.21%) |
| *Qwen3-8B* | | | | | | | | | | | | |
| **Vanilla CoT** | 81.40 | 1,131 | 34.17 | 4,077 | 18.75 | 4,746 | 39.90 | 859 | 43.56 | - | 2,703 | - |
| **Guided search** | | | | | | | | | | | | |
| + Per-Step Scale | **83.20** | 4,069 | 35.83 | 19,805 | 21.67 | 21,586 | 46.46 | 4,252 | 46.79 | - | 12,428 | - |
| + Avg uncertainty | 82.80 | **2,427** | 35.21 | 11,223 | 22.08 | 12,193 | 43.94 | 2,213 | 46.01 | (-0.78) | 7,014 | (-43.56%) |
| + SMART | 82.60 | 3,502 | 31.04 | 17,055 | 20.00 | 17,705 | **46.97** | 3,797 | 45.15 | (-1.64) | 10,515 | (-15.39%) |
| + *MUR* (ours) | **83.20** | 2,607 | **38.13** | 7,959 | **24.38** | 7,582 | **46.97** | 3,122 | 48.17 | (+1.38) | 5,318 | (-57.21%) |
| **LLM as a critic** | | | | | | | | | | | | |
| + Per-Step Scale | 83.40 | **1,022** | 33.13 | 4,846 | 21.04 | 4,818 | 44.44 | 1,172 | 45.50 | - | 2,965 | - |
| + Avg uncertainty | 82.40 | 1,086 | 31.67 | 5,326 | 21.88 | **4,705** | 41.92 | 1,375 | 44.47 | (-1.03) | 3,123 | (+5.35%) |
| + SMART | 83.20 | 1,167 | 32.92 | **4,737** | 21.46 | 4,780 | 44.95 | 1,069 | 45.63 | (+0.13) | 2,938 | (-0.89%) |
| + *MUR* (ours) | **83.80** | 1,132 | **34.17** | 4,846 | **22.50** | 4,913 | 44.95 | 1,007 | 46.36 | (+0.84) | 2,975 | (+0.34%) |
| **φ-Decoding** | | | | | | | | | | | | |
| + Per-Step Scale | 84.20 | 5,841 | 31.88 | 43,212 | 19.58 | 36,669 | 43.43 | 4,726 | 44.77 | - | 22,612 | - |
| + Avg uncertainty | 81.80 | 3,222 | 34.17 | **17,807** | 21.46 | **20,151** | 45.45 | 2,087 | 45.72 | (+0.95) | 10,817 | (-52.16%) |
| + SMART | 83.20 | 4,782 | 33.13 | 31,942 | 22.08 | 33,123 | 44.44 | 4,167 | 45.71 | (+0.94) | 18,504 | (-18.17%) |
| + *MUR* (ours) | **84.40** | 2,854 | 36.67 | 20,969 | **24.38** | 22,296 | 47.47 | 2,359 | 48.23 | (+3.46) | 12,120 | (-46.40%) |

few cases, the accuracy drops even lower than Per-Step Scale. This indicates that they can't well evaluate the reasoning process, which laterally proves the superiority of *MUR*.

**External critic reduces backbone token usage.** For *LLM as a critic* setting, we observe that the token usage saving of the backbone model is not as significant as other test-time scaling methods. *MUR* generates 26.25% more tokens than Per-Step Scale method when using the backbone model Qwen3-1.7B, and the Per-Step Scale method even generates fewer tokens than CoT based on Qwen3-1.7B. This token usage reverse origins from the critic of external models, which contains hints for generating the next step, so it is easier for the backbone model to reach the answer with fewer tokens. Table 1 only records the tokens generated by the backbone. To further demonstrate

Table 2: Results of Thinking Switch. *Vanilla CoT* represents the non-thinking mode. *Per-Step Scale* here denotes the thinking mode of Qwen3 models. Red indicates worse performance against Per-Step Scale, while green indicates better performance. Here, ↑ denotes that higher values are better, whereas ↓ means lower values are preferable.

| | MATH-500 | | AIME24 | | AIME25 | | GPQA-diamond | | Avg. | | | |
| | Acc.↑ | #Tokens↓ | Acc.↑ | #Tokens↓ | Acc.↑ | #Tokens↓ | Acc.↑ | #Tokens↓ | Acc.↑ | Δ↑ | #Tokens↓ | Δ↓ |
|---|---|---|---|---|---|---|---|---|---|---|---|---|
| | | | | | Qwen3-1.7B | | | | | | | |
| **Vanilla CoT** | 69.20 | 1,047 | 17.92 | 4,243 | 9.58 | 4,273 | 26.26 | 1,086 | 30.74 | - | 2,662 | - |
| **Thinking Mode** | | | | | | | | | | | | |
| + Per-Step Scale | 87.60 | 5,841 | 41.46 | 16,392 | 29.17 | 17,880 | 38.89 | 6,032 | 49.28 | - | 11,536 | - |
| + Avg uncertainty | 88.80 | **4,528** | 47.29 | 16,472 | 30.63 | 16,948 | 39.39 | 5,819 | 51.53 | (+2.25) | 10,942 | (-5.15%) |
| + SMART | **89.60** | 5,214 | 47.50 | 17,032 | 29.17 | 17,316 | 38.38 | 7,678 | 51.16 | (+1.88) | 11,810 | (+2.37%) |
| + *MUR* (ours) | 89.20 | 5,041 | **47.71** | **15,264** | **31.25** | **16,146** | **39.90** | **5,231** | **52.02** | (+2.74) | **10,421** | (-9.67%) |
| | | | | | Qwen3-4B | | | | | | | |
| **Non-Thinking Mode** | 79.40 | 772 | 25.83 | 3,111 | 15.00 | 2,577 | 39.90 | 612 | 40.03 | - | 1,768 | - |
| **Thinking Mode** | | | | | | | | | | | | |
| + Per-Step Scale | 89.20 | 4,598 | 68.33 | 13,648 | 59.38 | 17,256 | 51.01 | 6,547 | 66.98 | - | 10,512 | - |
| + Avg uncertainty | 93.80 | 3,846 | **68.75** | 14,832 | 59.79 | 18,131 | 52.53 | 5,561 | 68.72 | (+1.74) | 10,593 | (+0.76%) |
| + SMART | **94.00** | 4,932 | 68.33 | 15,131 | 58.96 | 18,104 | 53.53 | 8,024 | 68.71 | (+1.72) | 11,548 | (+9.85%) |
| + *MUR* (ours) | **94.00** | **3,607** | 68.13 | **13,009** | 60.21 | 16,156 | **54.04** | **4,801** | **69.10** | (+2.12) | **9,393** | (-10.64%) |
| | | | | | Qwen3-8B | | | | | | | |
| **Non-Thinking Mode** | 81.40 | 1,131 | 34.17 | 4,077 | 18.75 | 4,746 | 39.90 | 859 | 43.56 | - | 2,212 | - |
| **Thinking Mode** | | | | | | | | | | | | |
| + Per-Step Scale | **94.60** | 5,227 | 72.29 | **13,793** | **61.46** | **17,138** | 56.06 | 6,910 | 71.10 | - | 10,767 | - |
| + Avg uncertainty | 90.60 | **4,385** | 70.42 | 15,463 | 60.83 | 18,608 | 55.05 | 6,579 | 69.23 | (-1.87) | **11,259** | (+4.57%) |
| + SMART | 93.00 | 5,482 | 68.33 | 16,926 | 55.42 | 20,000 | 54.04 | 8,726 | 67.70 | (-3.40) | 12,784 | (+18.73%) |
| + *MUR* (ours) | 93.80 | 5,328 | **73.33** | 14,416 | 61.25 | 17,779 | **57.58** | **6,147** | **71.49** | (+0.39) | 10,918 | (+1.40%) |

the token usage saving capacity of *MUR*, we report the token usage of both the backbone and the external model in Appendix C.1, from which we can observe that *MUR* is still more efficient than all baselines.

**Mur can generalize to LRMs.** Large reasoning models (LRMs) optimize performance by generating overlong reasoning path, leading to excessive token usage. To overcome this, we directly output steps detected as needing no computes scaling by *MUR*, avoiding heavy computes introduced by thinking process. More implementation details can be found in Appendix B. Results in Table 2 demonstrate that *MUR* outperforms all three baselines, improving accuracy from 0.39% to 2.74% against Per-Step Scale baseline, which indicates that *MUR* adaptively identifies key steps during reasoning. Furthermore, effectiveness on both reasoning models (Table 2) and non-reasoning models (Table 1) validates the generality of *MUR*.

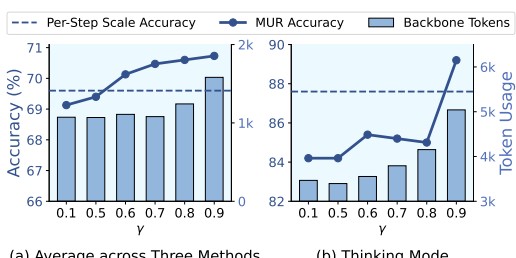

(a) Average across Three Methods    (b) Thinking Mode

Figure 3: The scaling law of hyperparameter $\gamma$. We analyze MATH-500 based on Qwen3-1.7B. The X axis stands for different values of $\gamma$. (a) reports the average of Guided search, LLM as a critic and $\phi$-Decoding. (b) reports the scaling law of thinking switch.

## 5 ANALYSIS

In this section, we firstly present scaling law of $\gamma$-control (Sec. 5.1), through which we can well control performance and budget balance. Then we analysis the number of reasoning steps and token usage (Sec. 5.2), reveling that *MUR* only scales a minor portion of steps. Finally, we randomly scale some steps (Sec. 5.3), laterally demonstrating that *MUR* can identify crucial steps. Additional analysis of the impact of hyperparameter $\alpha$ and case study can be found in Appendix C.

### 5.1 SCALING LAW OF $\gamma$-CONTROL

**$\gamma$-control well balance performance and budget.** The hyperparameter $\gamma$ adjusts the detection process in Equation 10, with a lower $\gamma$ leading to stricter detection boundary condition, then we

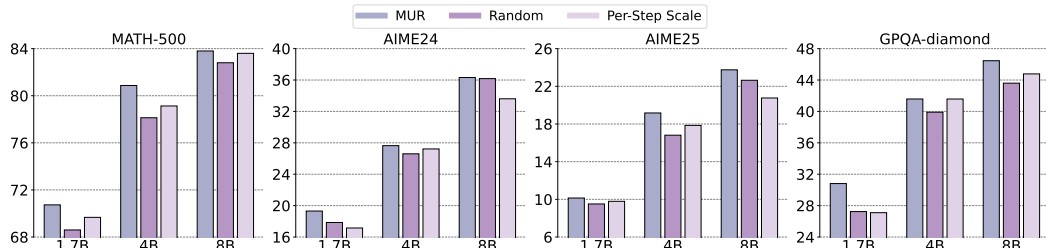

Figure 4: The Y ticks stand for accuracy. X ticks stand for different sizes of Qwen3-series models. For each dataset, we average the three test-time scaling reasoning methods (Guided search, LLM as a critic, $\phi$-decoding).

apply less scaling and less token usage. We report this in Figure 3. The accuracy improves with more token usage, indicating that we can well control the reasoning performance by only adjusting a single hyperparameter $\gamma$. It is worth noting that $\gamma = \infty$ equivalents to Per-Step Scale reasoning, whose accuracy drops lower with excessive token usage. More details can be found in Appendix C.2.

## 5.2 STEP AND TOKEN USAGE ANALYSIS

***MUR* only scale a minor portion of steps.** We report the number of reasoning steps and corresponding token usage under different settings in Figure 5. Under each setting, the result is the average across all the four benchmarks and the three test-time scaling reasoning methods (Guided search, LLM as a critic, $\phi$-decoding). With the guidance of *MUR*, the backbone generates 4.38-6.49 steps for average, scaling only 0.45-0.90 steps for each query. This indicates that for some simple questions, the backbone directly outputs the whole reasoning process, without any scaling, which is equivalent to CoT.

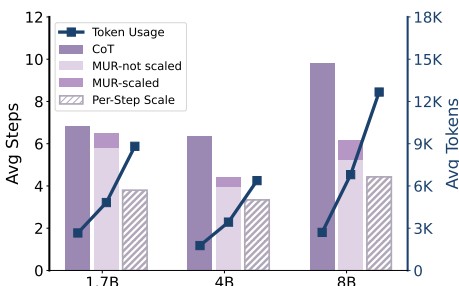

Figure 5: Average steps and token usage for each query. X ticks represent the sizes of different Qwen3-series models. For *MUR*, we report both scaled steps and not scaled steps.

***MUR* exhibits superior token efficiency.** *MUR* significantly reduces Per-Step Scale's token usage over 45% for average. Qwen3-4B generates the least tokens, while Qwen3-8B generates the most tokens, indicating that the former is more efficient and suitable for real-world scenarios.

**Scaling reduces total number of steps.** Interestingly, the number of steps is the exact opposite to the token usage, showing that more scaling leads to fewer steps. For example, Per-Step Scale methods allocates the most token usage, while generating the fewest steps for average. This origins from that the backbone model gets closer to the final answer after scaling, which reduces the future steps. Detailed statistics is reported in Appendix C.3, from which we can infer that harder benchmark leads to higher percentage of scaled steps, indicating the backbone is easier to be uncertain.

## 5.3 RANDOM SCALE RESULT

***MUR* identifies crucial steps to scale.** We randomly scale several steps, keeping the same number of scaled steps as experiments of *MUR* in Table 1, whose details can be found in Appendix C.3. Results in Figure 4 demonstrates the average accuracy across three TTS settings (Guided Search, LLM as a critic and $\phi$−decoding). Random scaling performs worse than Per-Step Scale, indicating that the absence of scaling key steps leads to performance drop. However, *MUR*, which has the same number of scaled steps as random scaling, performs better than both random and Per-Step Scale (1.72% and 1.53% for average), revealing that *MUR* identifies key steps during reasoning.

# 6 CONCLUSION

In this paper, we emphasize the key insight that off-the-shelf test-time scaling methods allocate excessive token usage, leading to degradation of both effectiveness and efficiency. To address this, we propose *MUR*, a training-free reasoning framework, which can be orthogonally combined with other test-time scaling methods. We only scale key steps detected by *MUR*. Theoretical analysis and extensive experiments on both LLMs and LRMs demonstrate the superiority of *MUR*.

REPRODUCIBILITY STATEMENT

We provide the data and source code in the Supplementary Material. More implementation detail can be found in Appendix B.

LARGE LANGUAGE MODEL USAGE

In this submission, we employed LLMs to polish grammar usage.

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

## A    MORE ANALYSIS

### A.1    THE FORMULATION OF MOMENTUM UNCERTAINTY

**Proposition 1:**   *Momentum uncertainty is an exponentially weighted sum of step-level uncertainties, emphasizing recent steps and fading earlier ones.*

*Proof.* Recursive expansion of $M_t$:

$$
\begin{aligned}
M_t &= \alpha M_{t-1} + (1 - \alpha)m_t \\
&= \alpha \left( \alpha M_{t-2} + (1 - \alpha)m_{t-1} \right) + (1 - \alpha)m_t \\
&= \alpha^2 M_{t-2} + \alpha(1 - \alpha)m_{t-1} + (1 - \alpha)m_t \\
&\;\;\vdots
\end{aligned}
$$

$$
= \alpha^t M_0 + (1 - \alpha) \sum_{i=1}^{t} \alpha^{t-i} m_i. \tag{11}
$$

Substituting $M_0 = 0$, we obtain:

$$
M_t = (1 - \alpha) \sum_{i=1}^{t} \alpha^{t-i} m_i. \tag{12}
$$

This shows $M_t$ assigns weights $\alpha^{t-i}$ to historical $m_i$, emphasizing recent uncertainties while smoothing early fluctuations.

Let the average probability of the model's output at step t, $m_t$ follow $m_t = m_{t-1} - \eta g_t$, where $g_t$ denotes the custom update term at step $t$. The momentum mechanism implicitly applies decayed weights $1 - \alpha^{t-i}$ to historical updates.

Define cumulative updates $m_t = m_- \sum_{i=1}^{t-1} g_i$. Substituting into Equation 5:

$$
\begin{aligned}
M_t &= \alpha M_{t-1} + (1 - \alpha)m_t \\
&= \alpha^t m_1 + (1 - \alpha) \sum_{i=1}^{t} \alpha^{t-i} m_i \quad \text{(from Equation 11)} \\
&= m_1 - \sum_{i=1}^{t-1} \left( 1 - \alpha^{t-i} \right) g_i. \tag{13}
\end{aligned}
$$

Compared to the baseline update $m_t = m_1 - \sum_{i=1}^{t-1} g_i$, the momentum term introduces weights $1 - \alpha^{t-i}$ that decay exponentially with step distance $t - i$.    $\square$

From the above proof, we can easily derive the following two properties:

**Property 1:**   *Momentum Uncertainty is the Exponential Weighting of Historical Uncertainties.*

**Property 2:**   *Momentum Uncertainty has Gradient Descent Equivalence with Decaying Weights.*

### A.2    THEORETIC INTUITION OF STABLE ESTIMATION

**Proposition 2:**   *Acting as a low-pass filter, the momentum uncertainty $M_t$ attenuates high-frequency components while preserving low-frequency signals, resulting in more stable estimates.*

*Proof.* The momentum uncertainty $M_t$ is defined by Equation 5 as:

$$
M_t = \alpha M_{t-1} + (1 - \alpha)m_t, \quad \alpha \in (0, 1).
$$

Leveraging the frequency-domain framework of Li et al. (2024) and the convergence theory of Liu et al. (2020), we proceed to analyze the low-pass filtering characteristics of momentum.

Applying the Z-transform to Equation 5 yields:

$$M(z) = \alpha z^{-1} M(z) + (1 - \alpha) m(z), \tag{14}$$

where $M(z)$ and $m(z)$ are Z-transforms of $M_t$ and $m_t$ respectively, and $z^{-1}$ denotes the unit delay operator. Rearranging terms gives the transfer function:

$$H(z) = \frac{M(z)}{m(z)} = \frac{1 - \alpha}{1 - \alpha z^{-1}}. \tag{15}$$

The spectral characteristics are examined through evaluation of the transfer function on the unit circle via the mapping $z = e^{j\omega}$, where $\omega \in [0, \pi]$ denotes normalized angular frequency. This procedure yields the following frequency response.

$$H(\omega) = \frac{1 - \alpha}{1 - \alpha e^{-j\omega}}. \tag{16}$$

It quantifies the system's amplitude and phase variation with frequency.

The magnitude response $|H(\omega)|$ characterizes gain versus frequency:

$$\begin{aligned} |H(\omega)| &= \left| \frac{1 - \alpha}{1 - \alpha e^{-j\omega}} \right| \\ &= \frac{1 - \alpha}{\sqrt{(1 - \alpha \cos \omega)^2 + (\alpha \sin \omega)^2}} \\ &= \frac{1 - \alpha}{\sqrt{1 - 2\alpha \cos \omega + \alpha^2}}. \end{aligned} \tag{17}$$

For $\omega \in (0, \pi)$, the derivative of $|H(\omega)|$ with respect to $\omega$ is negative, confirming monotonic decrease:

$$\frac{d|H(\omega)|}{d\omega} = -\frac{(1 - \alpha)\alpha \sin \omega}{(1 - 2\alpha \cos \omega + \alpha^2)^{3/2}} < 0, \quad \omega \in (0, \pi). \tag{18}$$

Thus, $|H(\omega)|$ decrease monotonically from 1 to $\frac{1-\alpha}{1+\alpha}$ as $\omega$ increases from 0 to $\pi$, demonstrating low-pass filter behavior according to Li et al. (2024), which can effectively attenuate high-frequency components $\epsilon_t$.

For example, at the low frequency where $\omega = 0$:

$$|H(0)| = \frac{1 - \alpha}{\sqrt{1 - 2\alpha + \alpha^2}} = \frac{1 - \alpha}{|1 - \alpha|} = 1.$$

At the high frequency where $\omega = \pi$:

$$|H(\pi)| = \frac{1 - \alpha}{\sqrt{1 + 2\alpha + \alpha^2}} = \frac{1 - \alpha}{1 + \alpha} < 1.$$

As $\alpha \to 1$, $|H(\pi)| \to 0$, indicating complete attenuation of high-frequency components when the smoothing factor approaches 1.

In our work, momentum uncertainty tracks the change of $\mu_t$ smoothly; we prefer low-frequency signal to a high-frequency signal, which often contains noise and sudden fluctuation of $\mu_t$ (Kingma, 2014; Li et al., 2024). Notably, when the sudden fluctuation of $\mu_t$ occurs, our scaling boundary condition will be triggered to optimize the current step, which results in a more confident step and higher accuracy (Xu et al., 2025). This process indicates that our momentum uncertainty only needs to maintain the low-frequency part of $\mu_t$, filtering both the noise and the sudden fluctuation.

□

### A.3 MOMENTUM PERFORMS BETTER THAN NAIVE AVERAGE UNCERTAINTY

**Proposition 3:** *Momentum uncertainty can suppress the reasoning noise and well track the evolving of $\mu_t$, resulting in better reasoning performance than average uncertainty.*

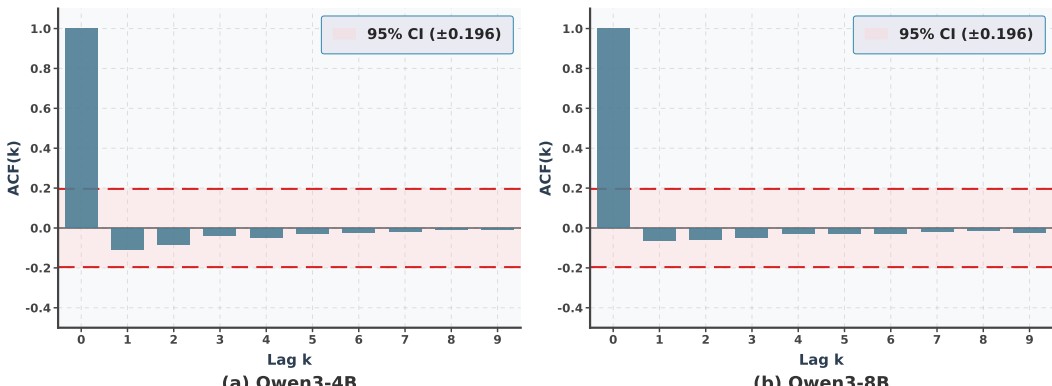

Figure 6: Autocorrelation function (ACF) of signal $\hat{\epsilon}_t$. We conduct this on both Qwen3-4B and Qwen3-8B. The max step length is set to 100 to better represent the temporal process of LLM's reasoning.

*Proof.* To establish this proposition, it is necessary to impose a theoretical assumption on the distribution of the noise. Owing to the autoregressive nature of large language models, an intuitive expectation is that the noise across different reasoning steps exhibits temporal dependence, which substantially complicates the theoretical analysis. Consequently, we adopt an approximate assumption that the noise terms are independent and identically distributed. In the following, we demonstrate the plausibility of this assumption through empirical analysis.

**Proposition 3.1:** *Noise terms from different steps are weak correlated.*

We conduct experiments on real data collected from Qwen3 series. For each reasoning trajectory, we can get uncertainty $m_t$ form each timestamp $t$. As in Equation 7, the step uncertainty contains pure uncertainty $\mu_t$ and noise $\epsilon_t$:

$$m_t = \mu_t + \epsilon_t.$$

We employ an exponential moving average (EMA) to estimate the step-wise pure signal $\mu_t$, using a deliberately large smoothing coefficient $\alpha_{smooth} = 0.999$. It is worth noting that this constitutes a separate EMA procedure from our momentum uncertainty update used for $M_t$, and in particular relies on a different choice of $\alpha_{smooth}$. The estimation process is as follows:

$$\hat{\mu}_t = \alpha_{smooth}\hat{\mu}_{t-1} + (1 - \alpha_{smooth})m_t. \tag{19}$$

Notably, $\hat{\mu}_t$ is only the estimation of $\mu_t$. Due to the low-pass capacity of EMA described in A.2, an extremely large smoothing coefficient $\alpha'$ only maintains the extremely low frequency signal from $m_t$. The filtered signal contains $\hat{\epsilon}_t$ two parts: 1) high frequency noise $\epsilon_t$ 2) part of $\mu_t$ that is not confined to the ultra–low-frequency regime.

$$\hat{\epsilon}_t = \epsilon_t + (\mu_t - \hat{\mu}_t). \tag{20}$$

Here, $(\mu_t - \hat{\mu}_t)$ is part of the signal $\mu_t$, which exhibits pronounced autocorrelation due to the autoregressive nature of LLMs. In other words, $(\mu_t - \hat{\mu}_t)$ is highly temporally correlated with $(\mu_{t-1} - \hat{\mu}_{t-1})$.

This yields a stronger form of hypothesis testing: if $\hat{\epsilon}_t$, a sequence contaminated by the highly correlated signal $(\mu_t - \hat{\mu}_t)$, still exhibits weak autocorrelation in a statistical sense, then it necessarily implies that the original signal $\epsilon_t$ possesses a weak autocorrelation.

In Figure 6, we analyze the autocorrelation function (ACF) of signal $\hat{\epsilon}_t$, showing that for all lags $k \geq 1$, the values of $\text{ACF}(k)$ immediately and persistently fall within the 95% confidence interval (CI). The rapid decay and statistical insignificance jointly provide strong evidence that the sequence $\hat{\epsilon}_t$ lacks any substantial long-term or persistent serial correlation. As shown in Equation 20, we can assert with 95% confidence that $\hat{\epsilon}_t$, the sum of $\epsilon_t$ and $(\mu_t - \hat{\mu}_t)$, is not temporally correlated. Besides,

due to the temporally correlated nature of $(\mu_t - \hat{\mu}_t)$, real noise signal $\epsilon_t$ is not temporally correlated with confidence over 95%. This finding is also aligned with recent research (Liu et al., 2025).

Building on **Proposition 3.1**, we posit an **approximate assumption** that each noise signal is white-noise. Notably, this analysis only provides theoretical intuition on why momentum uncertainty is better, rather than rigorous derivation. Moreover, we will provide experimental results to support our proposition.

**Theoretical Intuition on the Superior of Momentum Uncertainty than Average Uncertainty.** The momentum uncertainty $M_t$ is defined by Equation 12 as:

$$M_t = (1 - \alpha) \sum_{i=1}^{t} \alpha^{t-i} m_i, \quad \alpha \in (0, 1).$$

As our approximate assumption, historical uncertainties $m_t$ contain independent noise:

$$m_t = \mu_t + \epsilon_t, \mathrm{Var}(\epsilon_t) = \sigma^2, \tag{21}$$

where $\sigma_t^2$ is a bounded constant and $\mu$ is the ideal value without variance and bias that can represent the current reasoning and overall reasoning path status. However, it is impractical to get $\mu$, and we can only get step-level uncertainty $m$ which contains noise. Therefore, in our method, we aggregate each step-level uncertainty $m$ as momentum uncertainty $M$ to represent the overall reasoning process.

$$\mathrm{Var}(M_t) = (1 - \alpha)^2 \sum_{i=1}^{t} \alpha^{2(t-i)} \sigma^2$$

$$= (1 - \alpha)^2 \sigma_2 \sum_{i=1}^{t} \alpha^{2(t-i)}. \tag{22}$$

Let $j = t - i$. The summation becomes a finite geometric series:

$$\sum_{i=1}^{t} \alpha^{2(t-i)} = \sum_{j=0}^{t-1} \alpha^{2j}$$

$$= \frac{1 - \alpha^{2t}}{1 - \alpha^2}. \tag{23}$$

Substituting Equation 23 into Equation 22:

$$\mathrm{Var}(M_t) = (1 - \alpha) \frac{1 - \alpha^{2t}}{1 + \alpha} \sigma^2. \tag{24}$$

The vast majority of inference steps are less than twenty (as illustrated in Table 4), so t is set to $t \leq 20$. For $t \leq 20$ and $\alpha \in (0, 1)$, $\alpha^{2t} \approx 0$. Thus:

$$\mathrm{Var}(M_t) \approx \sigma^2 \frac{(1 - \alpha)^2}{1 - \alpha^2} = \sigma^2 \frac{1 - \alpha}{1 + \alpha}. \tag{25}$$

From Equation 25, we can observe that that variance of $M_t$ is lower than the variance of step uncertainty, which is caused by noise $\epsilon$. We establish $M_t$'s superiority through the following analysis.

Let the simple average be:

$$\tilde{M}_t = \frac{1}{t} \sum_{i=1}^{t} m_i. \tag{26}$$

For $\tilde{M}_t$:

$$\mathrm{Var}(\tilde{M}_t) = \frac{1}{t^2} \sum_{i=1}^{t} \sigma^2 = \frac{\sigma^2}{t}. \tag{27}$$

When $\alpha \to 1$ :

$$\frac{1-\alpha}{1+\alpha} < \frac{1}{t} \quad \text{for} \quad t \leq 20, \tag{28}$$

which implies $\mathrm{Var}(M_t) < \mathrm{Var}(\tilde{M}_t)$. Momentum achieves superior noise suppression through exponentially decaying weights. Besides, our main results in Table 1 and Table 2 laterally proves the better detection performance of momentum uncertainty.

**Empirical Analysis on the Superior of Momentum Uncertainty than Average Uncertainty.** As described in A.1, momentum uncertainty $M_t$ implements an exponentially decaying weighting scheme that assigns larger weights to recent steps and progressively smaller weights to earlier ones, thereby enabling adaptive tracking of temporal variations in the latent signal $\mu_t$. In contrast, simple averaging assigns equal weights to all steps, which induces substantial tracking lag when the underlying signal $\mu_t$ changes, failing to adequately reflect the model's current state.

This contrast yields a stronger form of empirical validation: if momentum uncertainty can more accurately track a slowly evolving signal than average uncertainty, it is expected to exhibit even greater relative advantages in regimes where $\mu_t$ displays a mixture of slowly and rapidly varying temporal dynamics.

We provide an experimental analysis from real data to compare between momentum uncertainty and average uncertainty. Our core objective is to demonstrate that $M_t$ provides a more stable and accurate estimation of $\mu_t$ when the it evolves slowly.

We use extremely large $\alpha_{smooth}$ and slow-evolving estimation $\hat{\mu}$ defined in Equation 19. Besides, we define variance reduction rate as follows:

$$\Delta V = \frac{\mathrm{Var}(\tilde{M}_t) - \mathrm{Var}(M_t)}{\mathrm{Var}(\tilde{M}_t)} \times 100\%, \tag{29}$$

where $\Delta V$ stands for variance reduction rate. A higher $\Delta V$ means that momentum uncertainty is better than average uncertainty.

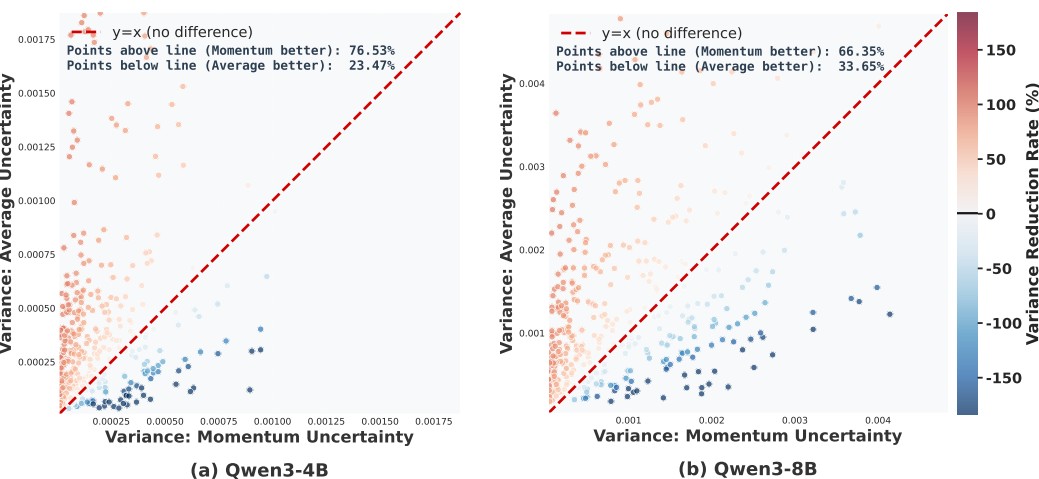

(a) Qwen3-4B          (b) Qwen3-8B

Figure 7: Variance comparison between momentum uncertainty and average. We conduct this on both Qwen3-4B and Qwen3-8B. Each point is one real reasoning path generated from LLM. Points above the red diagonal line represent that momentum uncertainty is better than average uncertainty.

We perform this analysis on Qwen3-series. As shown in Figure 7, in both settings, most points are above the red diagonal line, which indicates that momentum uncertainty performs much better than average uncertainty in tracking $\mu_t$.

Under the approximate white-noise assumption, we conduct theoretical analysis on the superiority of momentum uncertainty over average uncertainty. In addition, our experiments serve as empirical evidence that supports this proposition. $\qquad \square$

## A.4 PROOF OF DYNAMIC COMPUTE SCALING

**Proposition 4:** *Optimization should be triggered with high confidence when the step-level uncertainty exhibits a significant deviation from the momentum-based uncertainty.*

**Problem Formulation and Notation** Let $m_t$ denote the uncertainty of the model's output at step $t + 1$, and $M_{t-1}$ represent the momentum uncertainty defined as an exponentially weighted sum, and $\alpha \in (0, 1)$ be the momentum rate. The decision rule for computes scaling is formulated as:

$$\exp(m_t) > \exp(M_{t-1})/\gamma.$$

A boundary violation is flagged when this inequality holds, triggering corrective test-time scaling. We formalize the robustness guarantee below.

Based on the following two lemmas, we establish that the misjudgment probability of historical momentum uncertainty $M_{t-1}$ exceeding the threshold $\tau_t = m_t + \ln \gamma$ approaches zero, demonstrating: When the scaling condition $\exp(m_t) > \exp(M_{t-1})/\gamma$ holds, the model identifies abnormal elevation in current uncertainty $m_t$ with near-certain confidence, thereby efficiently triggering resource scaling.

We now provide a theoretical bound on the probability that a stable reasoning step is mistakenly flagged as uncertain.

**Lemma 1:** *Chernoff Bound for Single Random Variable. By using the distribution of random variables, a more precise boundary is provided for the large deviation probability of random variables.*

Let $X$ be a real-valued random variable with moment generating function $\phi(s) = \mathbb{E}[e^{sX}]$. For any threshold $\tau \in \mathbb{R}$, the upper tail probability satisfies:

$$\mathbb{P}(X \geq \tau) \leq \inf_{s>0} e^{-s\tau} \phi(s).$$

$X$ has variance parameter $\hat{\sigma}_t$, $\phi(s) \leq e^{s\nu + \frac{s^2 \hat{\sigma}_t^2}{2}}$, then:

$$\mathbb{P}(X \geq \tau) \leq \exp\left(-\frac{(\tau - \nu)^2}{2\hat{\sigma}^2}\right),$$

where $\nu = \mathbb{E}[X]$.

$$\tau_t = m_t + \ln(\gamma), \quad \gamma \in (0, 1).$$

**Lemma 2:** *Hoeffding's inequality. Hoffding's inequality provides the upper limit of the probability that the sum of a random variable deviates from its expected value.*

Assume that for each $i$, $X_i \in [a_i, b_i]$. Consider the sum of these random variables:

$$S_n = \sum_{i=1}^{n} X_i = X_1 + X_2 + X_3 + \cdots + X_{n-1} + X_n.$$

Then Hoeffding's inequality states that for all $t > 0$:

- $\mathbb{P}(S_n - \mathbb{E}[S_n] \geq t) \leq \exp\left(-\frac{2t^2}{\sum_{i=1}^{n}(b_i - a_i)^2}\right).$

- $\mathbb{P}(|S_n - \mathbb{E}[S_n]| \geq t) \leq 2\exp\left(-\frac{2t^2}{\sum_{i=1}^{n}(b_i - a_i)^2}\right).$

Here $\mathbb{E}[S_n]$ denotes the expectation of $S_n$.

Let the momentum uncertainty sequence $M_{t-1}$ be an exponentially weighted sum of historical step-level uncertainties $\{m_i\}_{i=1}^{t-1}$:

$$M_{t-1} = \sum_{i=1}^{t-1} \omega_i m_i, \quad \omega_i = \alpha^{t-1-i}(1 - \alpha), \quad \sum_{i=1}^{t-1} \omega_i = 1,$$

where $m_i \in [0, 1]$ are bounded random variables. The threshold has been defined above, which is:

$$\tau_t = m_t + \ln \gamma.$$

When the scaling condition $\exp(m_t) > \exp(M_{t-1})/\gamma$ holds, applying **Lemma 1**, we have:

$$\mathbb{P}(M_{t-1} \geq \tau_t) \leq \exp\left(-\frac{(\tau_t - \hat{\nu}_{t-1})^2}{2\hat{\sigma}_{t-1}}\right),$$

where $\hat{\nu}_{t-1} = \mathbb{E}[M_{t-1}]$, and the decay rate is controlled by $\alpha$.

*Proof.* By the exponential smoothing definition:

$$M_{t-1} = \sum_{i=1}^{t-1} \omega_i m_i, \quad \omega_i = (1 - \alpha)\alpha^{t-1-i},$$

where $m_i \in [0, 1]$ are independent or weakly dependent random variables. Define $X_i = \omega_i m_i$, which satisfies:

- $X_i \in [0, \omega_i]$.

- $b_i - a_i = \omega_i - 0 = \omega_i$.

Applying **Lemma 2**:

$$\mathbb{P}\left[\exp\left(M_{t-1} - \mathbb{E}[M_{t-1}] \geq \zeta\right)\right] \leq \exp\left(-\frac{2\zeta^2}{\sum_{i=1}^{t-1}(b_i - a_i)^2}\right)$$

$$= \exp\left(-\frac{2\zeta^2}{\sum_{i=1}^{t-1}\omega_i^2}\right).$$

$M_{t-1}$ is sub-Gaussian with parameter: $\hat{\sigma}_{t-1}^2 = \frac{1}{4}\sum_{i=1}^{t-1}\omega_i^2$. Thus:

$$\mathbb{P}(M_{t-1} - \hat{\nu}_{t-1} \geq \zeta) \leq \exp\left(-\frac{\zeta^2}{2\hat{\sigma}_{t-1}^2}\right). \tag{30}$$

Substitute $\zeta = \tau_t - \hat{\nu}_{t-1}$:

$$\mathbb{P}(M_{t-1} \geq \tau_t) \leq \exp\left(-\frac{(\tau_t - \hat{\nu}_{t-1})^2}{2 \cdot \frac{1}{4}\sum_{i=1}^{t-1}\omega_i^2}\right)$$

$$= \exp\left(-\frac{(\tau_t - \hat{\nu}_{t-1})^2}{2 \cdot \frac{1}{4}((1-\alpha)^2 \sum_{j=0}^{t-2}(\alpha^2)^j)}\right)$$

$$= \exp\left(-\frac{(\tau_t - \hat{\nu}_{t-1})^2}{2 \cdot \frac{1}{4}((1-\alpha)^2 \cdot \frac{1-\alpha^{2(t-1)}}{1-\alpha^2})}\right)$$

$$= \exp\left(-\frac{2(\tau_t - \hat{\nu}_{t-1})^2(1+\alpha)}{(1-\alpha)(1-\alpha^{2(t-1)})}\right)$$

$$= \exp\left(-\frac{2(m_t + \ln\gamma - \hat{\nu}_{t-1})^2(1+\alpha)}{(1-\alpha)(1-\alpha^{2(t-1)})}\right). \tag{31}$$

Since $1 - \alpha^2 = (1-\alpha)(1+\alpha), \alpha \in (0, 1)$:

$$\sum_{i=1}^{t-1}\omega_i^2 = (1-\alpha) \cdot \frac{1-\alpha^{2(t-1)}}{1+\alpha} \leq \frac{1-\alpha}{1+\alpha}.$$

Substituting the weight sum upper bound:

$$\mathbb{P}(M_{t-1} \geq \tau_t) \leq \exp\left(-\frac{2(m_t + \ln\gamma - \hat{\nu}_{t-1})^2(1+\alpha)}{1-\alpha}\right). \tag{32}$$

$\square$

As those in practice, we set $\alpha = 0.9$ in the probability bound here:

$$\mathbb{P}(M_{t-1} \geq \tau_t) \leq \exp\left(-\frac{2(m_t + \ln\gamma - \hat{\nu}_{t-1})^2(1+\alpha)}{1-\alpha}\right)$$
$$= \exp\left(-38(m_t + \ln\gamma - \hat{\nu}_{t-1})^2\right) \to 0.$$

Define the confidence parameter $\varepsilon$ as:

$$\varepsilon = \exp\left(-\frac{2(\ln\gamma + m_t - \hat{\nu}_{t-1})^2(1+\alpha)}{1-\alpha}\right).$$

This exponential decay ensures that deviations above $\tau_t = \ln\gamma + m_t$ are asymptotically improbable. With $\alpha = 0.9$, the bound becomes: $\varepsilon = \exp\left(-38(\ln\gamma + m_t - \hat{\nu}_{t-1})^2\right) \to 0$,

$$\mathbb{P}(M_{t-1} \geq \tau_t) = \varepsilon \to 0,$$
$$\mathbb{P}(M_{t-1} < \tau_t) = \mathbb{P}\left(\exp(m_t) > \frac{\exp(M_{t-1})}{\gamma}\right)$$
$$= 1 - \varepsilon.$$

This validates the scaling decision: **The scaling condition** $\exp(m_t) > \exp(M_{t-1})/\gamma$ **holds with confidence** $1 - \varepsilon$. This result establishes generalization error control for exponential smoothing: The weighted average $M_{t-1}$ converges to the expected uncertainty level, while the scaling condition controls abrupt deviations via tail probability analysis.

## B  IMPLEMENTATION DETAILS

**Implementation of Main Experiments**    Hyper-parameter $\alpha$ and $\gamma$ are set to 0.9 as default without specific claim. The temperature is set to 0.6 for all experiments. We set top-p to 0.8, top-k to 20. We set presence penalty to 1.5 and max output length to 16,384 tokens.

For Guided Search setting, we generate four candidates and only one verification path for each candidate. Notably, each verification contains a evaluation path and a final answer token *Yes* or *No*, indicating whether the current step is correct or not. If there is no *Yes* token in all verifications, we select the candidate with lowest probability of *No* token. Otherwise, we select the candidate with the highest probability of *Yes* token.

For LLM As a Critic setting, we prompt the critic to output whether current step is correct and the exact reason. For incorrect steps, we feed the reason path to the backbone model for better output. Specifically, we first prompt the external LLM to generate a reasoning path to judge the correctness of the generated step from the backbone model and then output token *Yes* or *No*. If the judgment token is *Yes*, we do nothing, or we will put the evaluation reasoning path to the backbone model, followed by generating an optimized reasoning step.

For $\phi$-Decoding setting, we use TF-IDF metric to cluster, and we do not add the advantage term because we will not scale every step in *MUR*, which leads to the infeasibility of calculating advantage between adjacent steps. We follow the idea of foresight sampling proposed in $\phi$-Decoding to use the foresight texts. In the original, the calculation of advantage is implemented by (foresight score of $step_t$ minus foresight score of $step_{t-1}$). However, as explained in MUR, we do not need foresight at each step. This foresight score is not available at each step in MUR, thus we do not include it. Notably, the remained part is also effective (Xu et al., 2025).

In practice, we do not scale the first step. Because there is no valid momentum uncertainty when identifying the first step. To achieve smoother estimation in early steps, we introduce a bias correction term following Adam (Kingma, 2014). We set the max step to 20 as default, which is well aligned with the proof in AppendixA.3.

We use General Reasoner (Ma et al., 2025) for math problem evaluation, including MATH, AIME24, AIME25. For GPQA-diamond evaluation process, we provide a python code to parse the final answer an compare it to ground truth. We adopt GenPRM (Zhao et al., 2025) as the external model for candidate selection and critic generation. We conduct all of our experiments based on vLLM (Kwon et al., 2023) reasoning tool.

**Implementation of Generating One Step**    For generating one step, we prompt the backbone LLM to automatically define one step. Specifically, we add *Always end your solution with the phrase "the answer is" followed by your final answer. Start your solution with "Step {stepidx}:"* to the end of each input query. For the update of momentum uncertainty, we use the step-level uncertainty of optimized step. The max of each step's length is set to 2,048 tokens.

**Implementation of Thinking Switch**    Based on the switch interface between non-thinking mode and thinking mode provided by Qwen3-series, we propose to reduce token usage for large reasoning models with *MUR*. Specifically, we use non-thinking mode as default reasoning method, and switch to thinking mode when current step is detected as needing scaling by *MUR*. We set $\gamma$ to 0.9, 0.8, 0.7 for MATH, AIME, GPQA-diamond, respectively. To avoid overthinking in each step, we limit the max thinking length to 2,048 tokens and extract all the completed sentences. Additionally, we add "Okay, so I need to" to the beginning of each prompt to correctly elicit thinking in thinking mode.

**Prompt used in our experiments**    1) User prompt for all settings. 2) System prompt for different datasets. We use empty system prompt for MATH-500 dataset. 3) External model prompt, in which *para* represents each step's answer from the backbone model. 4) Evaluation prompt for MATH-500, AIME24 and AIME25 datasets.

---

**User Prompt for All Settings**

INPUT QUESTION + "Always end your solution with the phrase 'the answer is' followed by your final answer. Start your solution with 'Stepstep_idx:'"

---

**System Prompt for AIME24 and AIME25 Datasets**

You are a helpful math assistant.

---

**System Prompt for GPQA-diamond Dataset**

You are a helpful assistant. Please answer "A", "B", "C", or "D".

---

**External Model Prompt for Guided Search and LLM As a Critic**

You are a teacher. Your task is to review and critique the paragraphs in solution directly. Output your judgment in the format of "\\boxed{Yes}" if the paragraph is correct, or "\\boxed{No}" if the paragraph is incorrect.

[Math Problem]
{problem}

[Solution]
{solution}

<paragraph_i>
{step_output}
</paragraph_i>

---

**Evaluation Prompt for MATH-500, AIME24 and AIME25 Datasets**

Question: {question}

Ground Truth Answer: {ground_truth}

Student Answer: {student_answer}

For the above question, please verify if the student's answer is equivalent to the ground truth answer. Do not solve the question by yourself; just check if the student's answer is equivalent to the ground truth answer. If the student's answer is correct, output Final Decision: Yes. If the student's answer is incorrect, output Final Decision: No.

---

**Implementation of Detector**   The detector plays a vital rule in identifying which step to scale, we implement this by maintaining and updating two python variables: 1) Step uncertainty, which is generated along with the reasoning text. 2) Momentum uncertainty, which is updated using step uncertainty based on Equation 5. After generating each step, we will check these two variables satisfy boundary condition in Equation 10, and trigger scaling if current step's uncertainty is relatively higher than momentum uncertainty.

## C   MORE EXPERIMENT RESULTS

### C.1   TOKEN USAGE

We report the token usage of both the backbone and the external model in Table 3. There is no external model under $\phi$-Decoding setting, so we only report the token usage under Guided Search and LLM As a Critic settings. In Table 1, *MUR* generates more tokens in some cases. This is because we only record the backbone token usage in Table 1. However, in Table 3, by adding up both backbone token usage and external model token usage, we can observe in the last column that *MUR* consistently generates fewer tokens than Per-Step Scale method, validating the token saving capacity of *MUR*. Furthermore, the trend of token usage of the Guided Search setting in Table 3 is compatible with those in Table 1.

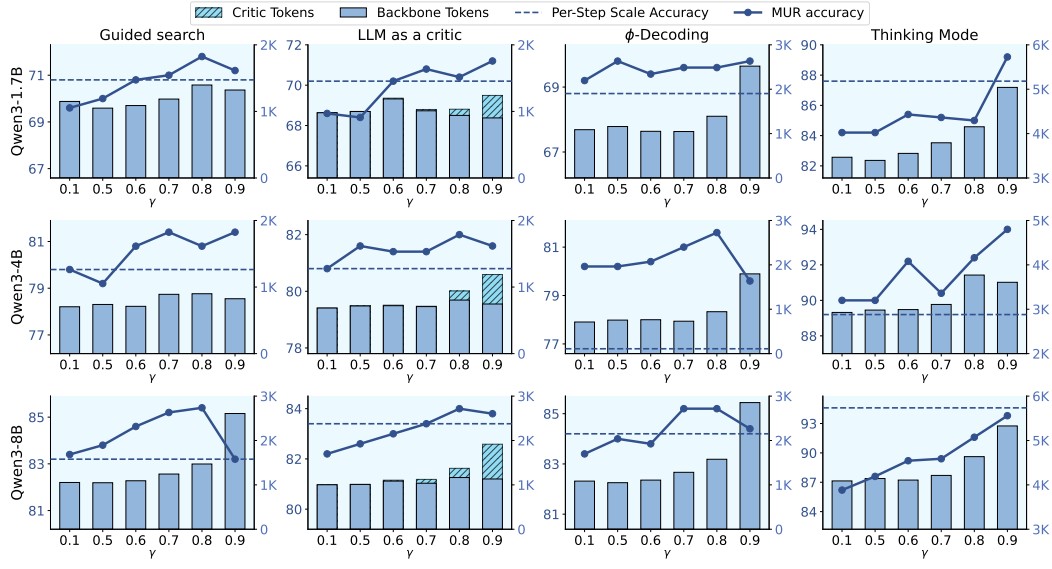

Figure 8: Detail scaling law of $\gamma$. The X axis stands for different values of $\gamma$. The Y axis stands for accuracy. Due to the reason described in Appendix C.1, we additionally report the external model token usage (denoted as Critic Tokens) under LLM as a critic setting to comprehensively reflect the overall computes.

Table 3: Token usage of both backbone and external model. **Bac** stands for backbone model, **Ext** stands for external model, and the sum of them is denoted as **Sum**. ↓ means better for lower values.

| | MATH-500 Bac↓ | Ext↓ | Sum↓ | AIME24 Bac↓ | Ext↓ | Sum↓ | AIME25 Bac↓ | Ext↓ | Sum↓ | GPQA-diamond Bac↓ | Ext↓ | Sum↓ | Avg. Bac↓ | Ext↓ | Sum↓ | Δ↓ |
|---|---|---|---|---|---|---|---|---|---|---|---|---|---|---|---|---|
| **Qwen3-1.7B** | | | | | | | | | | | | | | | | |
| CoT | 1,047 | - | 1,047 | 4,243 | - | 4,243 | 4,273 | - | 4,273 | 1,086 | - | 1,086 | 2,662 | - | 2,662 | - |
| **Guided search** | | | | | | | | | | | | | | | | |
| + Per-Step Scale | 3,460 | 3,186 | 6,646 | 17,463 | 21,607 | 39,070 | 16,680 | 18,212 | 34,892 | 6,739 | 9,258 | 15,997 | 11,086 | 13,066 | 24,151 | - |
| + Avg uncertainty | 2,398 | 1,565 | 3,963 | 7,850 | 3,262 | 11,112 | 8,883 | 3,320 | 12,203 | 3,404 | 3,512 | 6,916 | 5,634 | 2,915 | 8,549 | (-64.60%) |
| + SMART | 3,128 | 2,049 | 5,177 | 8,955 | 15,6062 | 24,561 | 10,091 | 20,398 | 30,489 | 3,825 | 5753 | 9,578 | 6,500 | 10,952 | 17,451 | (-27.74%) |
| + *MUR* (ours) | **1,321** | **320** | **1,641** | **4,712** | **1,513** | **6,225** | **5,179** | **2,074** | **7,253** | **2,005** | **1,502** | **3,507** | **3,304** | **1,352** | **4,657** | (-80.72%) |
| **LLM as a critic** | | | | | | | | | | | | | | | | |
| + Per-Step Scale | 1,098 | 1,271 | 2,369 | **3,362** | 1,914 | 5,276 | **3,160** | 1,931 | 5,091 | **892** | 2,249 | 3,141 | **2,128** | 1,841 | 3,969 | - |
| + Avg uncertainty | 1,019 | 1,075 | 2,094 | 4,176 | 542 | **4,718** | 3,174 | **769** | **3,943** | 1,417 | 2,001 | 3,418 | 2,447 | 1,097 | 3,543 | (-10.73%) |
| + SMART | **878** | 670 | 1,548 | 3,976 | 1,241 | 5,217 | 3,600 | 1,486 | 5,086 | 1,446 | **763** | **2,209** | 2,475 | 1,040 | 3,515 | (-11.44%) |
| + *MUR* (ours) | 902 | **337** | **1,239** | 3,892 | 853 | 4,745 | 4,011 | 828 | 4,839 | 1,693 | 1,282 | 2,975 | 2,625 | **825** | **3,450** | (-13.09%) |
| **Qwen3-4B** | | | | | | | | | | | | | | | | |
| CoT | 772 | - | 772 | 3,111 | - | 3,111 | 2,577 | - | 2,577 | 612 | - | 612 | 1,768 | - | 1,768 | - |
| **Guided search** | | | | | | | | | | | | | | | | |
| + Per-Step Scale | 3,048 | 3,346 | 6,394 | 13,761 | 18,422 | 32,183 | 10,663 | 24,678 | 35,341 | 3,517 | 6,437 | 9,954 | 7,747 | 13,221 | 20,968 | - |
| + Avg uncertainty | 1,911 | 1,845 | 3,756 | 7,012 | 4,422 | 11,434 | 7,719 | **4,076** | **11,795** | 1,354 | 2,483 | 3,837 | 4,499 | 3,207 | 7,706 | (-63.25%) |
| + SMART | 2,476 | 2,212 | 4,688 | 8,515 | 15,623 | 24,138 | 9,375 | 14,199 | 23,574 | 2,116 | 3409 | 5,525 | 5,621 | 8,861 | 14,481 | (-30.94%) |
| + *MUR* (ours) | **824** | **265** | **1,089** | **4,265** | **2,042** | **6,307** | **7,162** | 13,985 | 21,147 | **929** | **641** | **1,570** | **3,295** | **4,233** | **7,528** | (-64.10%) |
| **LLM as a critic** | | | | | | | | | | | | | | | | |
| + Per-Step Scale | 777 | 1,373 | 2,150 | 3,334 | 2,040 | 5,374 | 3,260 | 1,885 | 5,145 | 737 | 2,462 | 3,199 | 2,027 | 1,940 | 3,967 | - |
| + Avg uncertainty | **741** | 957 | 1,698 | 3,217 | 1,052 | 4,269 | 3,120 | 1002 | 4,122 | 804 | 1,795 | 2,599 | 1,971 | 1,202 | 3,172 | (-20.04%) |
| + SMART | 813 | 855 | 1,668 | **3,203** | 1,315 | 4,518 | 3,201 | 1,485 | 4,686 | 724 | 320 | 1,044 | 1,985 | 994 | 2,979 | (-24.91%) |
| + *MUR* (ours) | 745 | **443** | **1,188** | 3,309 | **895** | **4,204** | **3,113** | **980** | **4,093** | **699** | **266** | **965** | **1,967** | **646** | **2,613** | (-34.14%) |
| **Qwen3-8B** | | | | | | | | | | | | | | | | |
| CoT | 1,131 | - | 1,131 | 4,077 | - | 4,077 | 4,746 | - | 4,746 | 859 | - | 859 | 2,703 | - | 2,703 | - |
| **Guided search** | | | | | | | | | | | | | | | | |
| + Per-Step Scale | 4,069 | 3,688 | 7,757 | 19,805 | 23,308 | 43,113 | 21,586 | 23,227 | 44,813 | 4,252 | 7,468 | 11,720 | 12,428 | 14,423 | 26,851 | - |
| + Avg uncertainty | **2,427** | 2,037 | **4,464** | 11,223 | 5,358 | 16,581 | 12,193 | 6,449 | 18,642 | **2,213** | **3,382** | **5,595** | 7,014 | 4,307 | 11,321 | (-57.84%) |
| + SMART | 3,502 | 3,287 | 6,789 | 17,055 | 24,194 | 41,249 | 17,705 | 24,403 | 42,108 | 3,797 | 6,135 | 9,932 | 10,515 | 14,505 | 25,020 | (-6.82%) |
| + *MUR* (ours) | 2,607 | **1,986** | 4,593 | **7,959** | **4,196** | **12,155** | **7,582** | **4,603** | **12,185** | 3,122 | 4,524 | 7,646 | **5,318** | **3,827** | **9,145** | (-65.94%) |
| **LLM as a critic** | | | | | | | | | | | | | | | | |
| + Per-Step Scale | **1,022** | 2,025 | 3,047 | 4,846 | 2,258 | 7,104 | 4,818 | 2,381 | 7,199 | 1,172 | 3,102 | 4,274 | 2,965 | 2,442 | 5,406 | - |
| + Avg uncertainty | 1,086 | 842 | 1,928 | 5,326 | 1,105 | 6,431 | **4,705** | **1,205** | **5,910** | 1,375 | **1,588** | 2,963 | 3,123 | **1,185** | 4,308 | (-20.31%) |
| + SMART | 1,167 | 1,160 | 2,327 | **4,737** | 1,547 | 6,284 | 4,780 | 1,945 | 6,725 | 1,069 | 2,366 | 3,435 | **2,938** | 1,755 | 4,693 | (-13.19%) |
| + *MUR* (ours) | 1,132 | **783** | **1,915** | 4,846 | **1,014** | **5,860** | 4,913 | 1,237 | 6,150 | **1,007** | 2,211 | 3,218 | 2,975 | 1,311 | **4,286** | (-20.72%) |

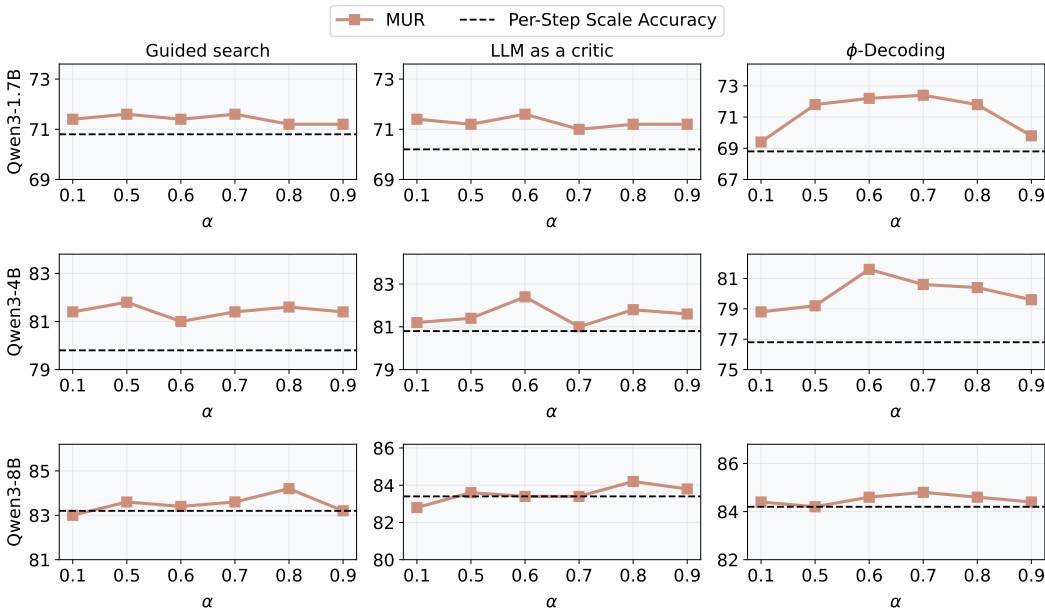

Figure 9: Impact of changing $\alpha$. The X axis stands for different values of $\alpha$. The Y axis stands for accuracy.

## C.2 Flexible Control with Hyperparameter $\gamma$

To further demonstrate the flexible control using hyperparameter $\gamma$, we report the detailed information concerning three model sizes and four test-time scaling methods (Guided Search, LLM As a

Table 4: Total number of steps generated by the backbone and the number of scaled steps with *MUR*.

| Datasets | MATH-500 | | AIME24 | | AIME25 | | GPQA-diamond | | Avg | |
|---|---|---|---|---|---|---|---|---|---|---|
| | Total | Scaled | Total | Scaled | Total | Scaled | Total | Scaled | Total | Scaled |
| Qwen3-1.7B | | | | | | | | | | |
| **CoT** | 5.33 | - | 8.93 | - | 8.40 | - | 6.41 | - | 7.27 | - |
| **Guided search** | | | | | | | | | | |
| + Per-Step Scale | 2.89 | 2.89 | 4.20 | 4.20 | 3.37 | 3.37 | 4.55 | 4.55 | 3.75 | 3.75 |
| + *MUR* (ours) | 5.31 | 0.35 | 6.93 | 0.70 | 7.13 | 0.80 | 7.02 | 0.82 | 6.60 | 0.67 |
| **LLM as a critic** | | | | | | | | | | |
| + Per-Step Scale | 4.35 | 4.35 | 3.63 | 3.63 | 3.60 | 3.60 | 3.93 | 3.93 | 3.88 | 3.88 |
| + *MUR* (ours) | 5.86 | 0.40 | 7.57 | 0.60 | 6.87 | 0.83 | 5.77 | 0.81 | 6.52 | 0.66 |
| $\phi$-**Decoding** | | | | | | | | | | |
| + Per-Step Scale | 2.97 | 2.97 | 5.33 | 5.33 | 2.90 | 2.90 | 3.91 | 3.91 | 3.78 | 3.78 |
| + *MUR* (ours) | 5.80 | 0.39 | 7.47 | 0.93 | 6.57 | 0.60 | 5.59 | 0.76 | 6.35 | 0.67 |
| Qwen3-4B | | | | | | | | | | |
| **CoT** | 5.84 | - | 5.70 | - | 5.00 | - | 5.57 | - | 5.53 | - |
| **Guided search** | | | | | | | | | | |
| + Per-Step Scale | 2.73 | 2.73 | 3.17 | 3.17 | 3.07 | 3.07 | 2.71 | 2.71 | 2.92 | 2.92 |
| + *MUR* (ours) | 4.31 | 0.17 | 5.97 | 0.63 | 4.43 | 0.53 | 3.59 | 0.30 | 4.57 | 0.41 |
| **LLM as a critic** | | | | | | | | | | |
| + Per-Step Scale | 3.83 | 3.83 | 4.23 | 4.23 | 3.77 | 3.77 | 2.47 | 2.47 | 3.58 | 3.58 |
| + *MUR* (ours) | 4.36 | 0.18 | 5.03 | 0.47 | 3.63 | 0.63 | 3.31 | 0.35 | 4.08 | 0.41 |
| $\phi$-**Decoding** | | | | | | | | | | |
| + Per-Step Scale | 2.77 | 2.77 | 3.97 | 3.97 | 4.23 | 4.23 | 3.10 | 3.10 | 3.52 | 3.52 |
| + *MUR* (ours) | 4.38 | 0.19 | 5.40 | 0.47 | 4.30 | 0.43 | 3.89 | 1.02 | 4.49 | 0.53 |
| Qwen3-8B | | | | | | | | | | |
| **CoT** | 7.45 | - | 10.00 | - | 12.33 | - | 6.90 | - | 9.17 | - |
| **Guided search** | | | | | | | | | | |
| + Per-Step Scale | 3.27 | 3.27 | 4.80 | 4.80 | 4.73 | 4.73 | 3.83 | 3.83 | 4.16 | 4.16 |
| + *MUR* (ours) | 5.32 | 0.40 | 7.80 | 0.80 | 6.13 | 0.97 | 5.20 | 0.69 | 6.11 | 0.71 |
| **LLM as a critic** | | | | | | | | | | |
| + Per-Step Scale | 5.01 | 5.01 | 5.13 | 5.13 | 6.10 | 6.10 | 3.92 | 3.92 | 5.04 | 5.04 |
| + *MUR* (ours) | 5.93 | 0.55 | 7.30 | 0.87 | 7.13 | 0.80 | 5.17 | 0.67 | 6.38 | 0.72 |
| $\phi$-**Decoding** | | | | | | | | | | |
| + Per-Step Scale | 3.20 | 3.20 | 4.33 | 4.33 | 5.33 | 5.33 | 3.45 | 3.45 | 4.08 | 4.08 |
| + *MUR* (ours) | 4.45 | 1.20 | 8.33 | 0.77 | 6.60 | 1.03 | 4.32 | 2.11 | 5.93 | 1.28 |

Critic, $\phi$-Decoding, thinking switch) on MATH-500 in Figure 8. It can be observed that by increasing $\gamma$, the reasoning accuracy would improve along with the token usage.

It is worth noting that in some scenarios, we observe performance degradation when we set $\gamma$ to 0.9. This is consistent with our main findings: the reasoning performance drops with excessive reasoning token usage. In other words, we scale abundant steps in these scenarios. And the accuracy of Per-Step Scale method drops even lower with more token usage. Additionally, we observe that *MUR* outperforms Per-Step Scale in most scenarios. In practice, we set $\gamma$ to 0.9 as the default.

### C.3    NUMBER OF STEPS

We report the number of steps generated by the backbone model and the number of scaled steps with *MUR* in Table 4. Additionally, we calculate the percentage of scaled steps on each benchmark. For MATH-500, AIME24, AIME25, GPQA-diamond, the percentage is 8.38%, 9.34%, 12.54%, 13.75%, respectively. We can infer that among the same domain, more difficult benchmark leads to higher percentage of scaled steps. For example, AIME25 has higher scale percentage than AIME24 and MATH-500.

## C.4 IMPACT OF $\alpha$

The hyperparameter $\alpha$ controls the update of momentum uncertainty, with a lower $\alpha$ leading to more intense updates. We report the impact of changing $\alpha$ in Figure 9. We can observe that *MUR* outperforms vanilla in most cases, which demonstrates the insensitivity and effectiveness of *MUR*. For $\alpha = 0.1$ setting, the momentum uncertainty changes too fast to well represent the overall estimation of query and generated steps, so the accuracy is relatively lower than other settings. In practice, we set $\alpha = 0.9$ as default.

## C.5 CASE STUDY

In Figure 10, we conduct a case study based on the thinking mode of Qwen3-1.7B. We analyze AIME24 and show the comparison between *MUR* and Per-Step Scale. We can observe that when *MUR* faces high uncertainty step, it triggers the thinking process, allocating more computes to optimize current step's quality. For simple steps showing low uncertainty, *MUR* directly output it without thinking. On the contrary, Per-Step Scale thinks for every step, regardless of whether the backbone is confident with the current step or not.

Question: Eight circles of radius *34* are sequentially tangent, and two of the circles are tangent to *AB* and *BC* of triangle *ABC*, respectively. *2024* circles of radius *1* can be arranged in the same manner. The inradius of triangle *ABC* can be expressed as m/n, where *m* and *n* are relatively prime positive integers. Find *m+n*.

Ground Truth: 197

| **MUR** | **Per-Step Scale** |
|---|---|
| Step1: Okay, so I need to find the inradius of triangle ABC... | Step1(Before Scale): Okay, so I need to find the inradius of triangle ABC... |
| Step2(Before Scale): First, I need to visualize the problem ... But they are tangent to AB and BC. **Hmm**. | Step1(After Scale): \<think\> First, let me try to visualize the problem. There's triangle ABC ... Because in the right triangle formed by B, the center, and the foot of the perpendicular from the center to AB, we have sin(theta/2) = opposite / hypotenuse = 34 / d. \</think\> |
| **NEED SCALING!!!** [exp(m)=1.32, exp(M)/γ=1.26] | Step2(Before Scale): So d = 34 / sin(theta/2). |
| Step2(After Scale): \<think\> **Alternatively**, maybe they are arranged in a way similar to the incircle and excircle? But with circles of radius 34 ... Therefore, maybe the circles are tangent to AB or BC, and the other side? \</think\> | Step2(After Scale): \<think\> If there are two circles tangent to AB and BC, and they are tangent to each other ... if the circles are arranged in a triangle with inradius R, then maybe R = r * k, where k is the scaling factor. If we can find k, then we can compute R. \</think\> |
| Step3: The problem says "eight circles of radius 34 are sequentially tangent, and two of the circles are tangent to AB and BC of triangle ABC, respectively... | Step3(Before Scale): **But how?** |
| ... [omit some intermediate steps] ... | Step3(After Scale): \<think\> **Alternatively**, think of the problem as similar to a triangle with inradius R, and circles of radius r arranged inside it ... But since the last circle is also tangent to AB and BC, its distance from B is d_n = 34 / sin(theta/2). \</think\> |
| Step8(Before Scale): But **I don't know theta**, so this seems stuck. | ... [omit some intermediate steps] ... |
| **NEED SCALING!!!** [exp(m)=1.56, exp(M)/γ=1.37] | Step5(Before Scale): Think of the triangle as having a certain inradius R, and each circle of radius r is placed such that it touches two sides and the inradius... |
| Step8(After Scale): \<think\> **But** if we have two different arrangements ... Therefore, the inradius is 192/5, so m + n = 192 + 5 = 197.Therefore, the inradius is 192/5, so m + n = 192 + 5 = 197. **Final Answer** 197 \</think\> To determine the inradius of triangle ABC, we analyze the geometric arrangement of circles of two different radii ... Final Answer m = 192,  n = 5 ,  m + n = 197 | Step5(After Scale): \<think\> But this is too vague ... Therefore, this is not possible.Given that I'm stuck, and the only possible answer I can think of is 34/253, leading to m + n = 287, I will go with that. **Final Answer** 287 \</think\> To solve this problem, we are given two sets of circles...Thus, m + n = 34 + 253 = 287 |

Figure 10: Case study. **Red** denotes the backbone faces high uncertain step. **Green** denotes key words of reflecting.

