# OpenReview forum: "MUR: Momentum Uncertainty guided Reasoning for Large Language Models"
_ICLR.cc/2026/Conference — ICLR 2026 Conference Withdrawn Submission_

### Official Review · Reviewer_iSh3 · 2025-10-26

**Soundness:** 3
**Presentation:** 3
**Contribution:** 2
**Rating:** 4
**Confidence:** 3

**Summary:**

This paper addresses the issue of "overthinking" in Large Language Models (LLMs), where Test-Time Scaling (TTS) methods waste computation on simple reasoning steps. The authors propose Momentum Uncertainty-guided Reasoning (MUR), a training-free framework to adaptively apply TTS. The authors claim this method reduces computation by over 45% while improving accuracy on several reasoning benchmarks.

**Strengths:**

1. The proposed method is training-free and designed to be an orthogonal framework that can be combined with existing TTS methods, which is a good quality for a utility-focused method.
2. Efficiency is an important issue  for TTS

**Weaknesses:**

1. Text Entropy may not to be a good metric for confidence.
2. I am not sure whether the simple baseline like randomly selecting steps to refine or selecting the first or the last few steps would be better or comparable with MUR.

**Questions:**

1. Why not consider the adaptive thinking on scaling the number of steps
2. How do you identify the intermediate steps

---

> ### Author Response · Authors · 2025-11-20
> **Response to Reviewer iSh3**
>
> Thanks for your support in our generalizable method, practical application and motivations.
>
> > Concern 4-1: Text Entropy may not be a good metric for confidence.
>
> Thanks for pointing out your concern. First of all, our paper mainly focuses on how to utilize uncertainty (i.e., text entropy) to help dynamic scaling, rather than propose a superior approach to evaluate confidence. Second, the use of text entropy in our paper is motivated by some recent paper [1,2] to represent confidence. They demonstrate the effectiveness of this simple confidence metric.
>
> [1]Confidence Improves Self-Consistency in LLMs
>
> [2]Evaluating Log-Likelihood for Confidence Estimation in LLM-Based Multiple-Choice Question Answering
>
>
> > Concern 4-2: I am not sure whether the simple baseline like randomly selecting steps to refine or selecting the first or the last few steps would be better or comparable with MUR.
>
> Thanks for raising the question. We have included the experiments of random selection in **Section 5.3** of the original paper. The results reveal that this baseline fail to identify the proper steps, leading to obvious performance drops.
>
> Also, based on your question, we also supplement the experiments of scaling the first and last step. The comparison results (Accuracy, \#Tokens)are as follows:
>
> **Qwen3-1.7B**
>
> | Method | Variant | MATH | | AIME24 | | AIME25 | | GPQA-D | | Avg. | |
> | :--- | :--- | :--- | :--- | :--- | :--- | :--- | :--- | :--- | :--- | :--- | :--- |
> | **Guided Search** | Scale First Step | 70.40 | 1,234 | 16.67 | 4,647 | 10.00 | 5,321 | 23.23 | 1,460 | 30.08 | 3,166 |
> | | Scale Last Step | 73.40 | 2,049 | 16.67 | 6,790 | 10.00 | 6,879 | 24.24 | 1,527 | 31.08 | 4,311 |
> | | MUR | 71.20 | 1321 | 18.33 | 4,712 | 10.63 | 5,179 | 32.83 | 2,005 | 33.25 | 3,304 |
>
> **Qwen3-4B**
>
> | Method | Variant | MATH | | AIME24 | | AIME25 | | GPQA-D | | Avg. | |
> | :--- | :--- | :--- | :--- | :--- | :--- | :--- | :--- | :--- | :--- | :--- | :--- |
> | **Guided Search** | Scale First Step | 81.80 | 1,015 | 20.00 | 2,951 | 10.00 | 3,712 | 41.41 | 866 | 38.30 | 2,136 |
> | | Scale Last Step | 83.00 | 1,541 | 20.00 | 6,159 | 16.67 | 6,304 | 36.87 | 1,035 | 39.14 | 3,760 |
> | | MUR | 81.40 | 824 | 29.58 | 4,265 | 19.17 | 7,162 | 41.92 | 929 | 43.02 | 3,295 |
>
> From the results, we can observe that, these settings can not dynamically choose which step to optimize, thus leading to sub-optimal results.
>
> > Concern 4-3: Why not consider the adaptive thinking on scaling the number of steps
>
> Under our setting, LLMs are encouraged to reason step by step, naturally leading them to produce as many intermediate steps as possible. Therefore, this kind of scaling is inherently considered.
>
> > Concern 4-4: How do you identify the intermediate steps ?
>
> We have included the details in Appendix B of the original paper. We leverage the LLM itself to implement it, prompting the LLM to explicity generated a step at once, using “Step {stepidx}:”  to seperate each step. Please refer to **Appendix B** for detail.

---

### Official Review · Reviewer_vwTG · 2025-10-31

**Soundness:** 3
**Presentation:** 2
**Contribution:** 2
**Rating:** 4
**Confidence:** 3

**Summary:**

This paper proposes an adaptive inference-budget allocation scheme for Test-Time Scaling (TTS). By aggregating uncertainty through a designed binary detector, the proposed method effectively reduces the number of reasoning tokens on multiple datasets without sacrificing accuracy.

**Strengths:**

1) The proposed scheme significantly reduces inference token consumption without sacrificing accuracy.
2) The authors provide theoretical analyses that motivate their design, helping readers to better understand the underlying principles of the method.

**Weaknesses:**

1) On the theoretical side, the assumptions are overly strong. The authors should at least provide evidence that the prerequisites required by their proofs hold, or verify them experimentally. For example, Equation (7) models the fluctuations in step-level uncertainty using an independent white noise model; however, for autoregressive language models, independence clearly does not hold. Even if a simplified model is used for analysis, the authors should provide toy examples that satisfy the assumptions to demonstrate their validity. In the proof of Proposition 3, the authors “assume μ_i converges exponentially to μ_∞ as timestamp increases,” which is also a very strong assumption that is neither validated nor supported by references. Moreover, all theorems should ideally be verified under restricted conditions through carefully designed toy experiments before being extended to the LLM context. Otherwise, the theoretical analysis and the method design remain disconnected, and it is impossible to determine whether the analytical conclusions actually support the method’s effectiveness.

2) On the experimental side, although the proposed method improves inference efficiency, the additional inference latency introduced by the external detector should also be compared. The authors should include a comparison of the actual inference time across different methods.

3) Furthermore, the authors provide very little description of their component designs (e.g., the specific design of the detector) and experimental details (e.g., evaluation protocols for each task, inference parameter settings), which significantly limits the reproducibility of the paper.

**Questions:**

Overall, I find the motivation of this work very interesting. However, in its current form it does not appear to be sufficiently prepared.

---

> ### Author Response · Authors · 2025-11-20
> **Response to Reviewer vwTG (Part 1/2)**
>
> Thanks for your support in our performances and theoretical motivations.
>
> > Concern 3-1: On the theoretical side, the assumptions are overly strong.
>
> Thanks for raising your concern and giving valuable suggestions.
>
> (1) **On the validity of Gaussian noise assumption.**
>
> We acknowledge that the Gaussian noise assumption is a simplification due to the inherent difficulty in precisely modeling the complex noise distribution in LLM generation. To enhance the rigor of this assumption, we supplemented our analysis with empirical evidence. We perform an autocorrelation function(ACF) experiment on real data generated from the Qwen3 series, showing that **we have confidence over 95% that the noise is not temporally correlated**. Based on this empirical finding, the assumption of Gaussian noise becomes a more justifiable and theoretically tractable approximation, which allows us to provide the core theoretical intuition behind the superiority of MUR. This finding also aligns with recent work[1] assumption. Details can be found in **Appendix A.3** in the new version.
>
> (2) **On the assumption "μ_i converges exponentially to μ_∞ as timestamp increases"**
>
> Based on your suggestion, we adjust this theoretical part to avoid the simplified assumption. To support the proof of superiority, we further provide the theoretical proof from a low-pass filter perspective[2]. Momentum serves as a low-pass filter which removes the high-frequency noise and sudden fluctuation of uncertainty, so that we can smoothly and dynamically track the evolution of reasoning uncertainty. Details can be found in **Sec 3.2** and **Appendix A.2** in the new version.
>
> (3) **Bridging the theoretical analysis and the actual reasoning performance.**
>
> The updated theoretical analysis is to strengthen the motivation and support the superior properties of Momentum Uncertainty. These advantages in theory are mainly reflected in the performances over various baselines in Table 1. The results demonstrate that MUR is a robust and excellent choice among various benchmarks, corresponding to the proof that "Momentum uncertainty removes high-frequency noise and sudden fluctuation of uncertainty".
>
> Notably, the updated **Appendix A.3** presents the comparison with naive averaging baseline, from both the theoretical and empirical perspective. It can bridge the gap between theoretical analysis and actual reasoning performance gains.
> Additionally, the proof of dynamic compute scaling in **Appendix A.5** can be empirically verified by the thinking switch experiments in **Sec. 5.1** and **Appendix C.2**.
>
> [1] Large language models and causal inference in collaboration: A comprehensive survey
>
> [2] On the Performance Analysis of Momentum Method: A Frequency Domain Perspective

---

> ### Author Response · Authors · 2025-11-20
> **Response to Reviewer vwTG (Part 2/2)**
>
> > Concern 3-2: On the experimental side, include a comparison of the actual inference time across different methods.
>
> Thank you for your suggestion. The external detector's implementation is based on several lines of code maintining two float python variables, only requiring CPU computation, so its processing time is negligible. As you recommended, we have now included a comparison of the actual inference time (s) across all baselines.
>
> **Qwen3-1.7B**
>
> | Method | Variant | MATH | AIME24 | AIME25 | GPQA-diamond | Avg. |
> | :--- | :--- | :--- | :--- | :--- | :--- | :--- |
> | **CoT** | | 3.94 | 15.58 | 15.69 | 4.43 | 9.91 |
> | **Guided Search** | Per-Step Scale | 12.80 | 92.47 | 73.34 | 31.82 | 52.61 |
> | | Avg. Uncertainty | 8.68 | 31.67 | 31.67 | 17.62 | 22.41 |
> | | SMART | 10.57 | 62.77 | 55.03 | 21.11 | 37.37 |
> | | MUR | 4.96 | 27.25 | 25.11 | 9.89 | **16.80** |
> | **LLM As a Critic** | Per-Step Scale | 8.35 | 23.24 | 19.62 | 10.63 | **15.46** |
> | | Avg. Uncertainty | 7.32 | 27.19 | 21.13 | 12.11 | 16.94 |
> | | SMART | 5.38 | 48.2 | 44.93 | 8.14 | 26.66 |
> | | MUR | 4.49 | 30.16 | 25.4 | 10.96 | 17.75 |
> | **$\phi$-Decoding** | Per-Step Scale | 15.52 | 65.28 | 62.56 | 30.14 | 43.38 |
> | | Avg. Uncertainty | 9.24 | 36.33 | 38.61 | 10.03 | 23.55 |
> | | SMART | 11.26 | 62.73 | 50.21 | 13.61 | 34.45 |
> | | MUR | 8.12 | 35.21 | 36.6 | 7.29 | **21.81** |
>
>
> **Qwen3-4B**
>
> | Method | Variant | MATH | AIME24 | AIME25 | GPQA-D | Avg. |
> | :--- | :--- | :--- | :--- | :--- | :--- | :--- |
> | **CoT** | | 9.46 | 22.6 | 21.9 | 8.57 | 15.63 |
> | **Guided Search** | Per-Step Scale | 31.12 | 134.4 | 125.8 | 86.06 | 94.35 |
> | | Avg. Uncertainty | 21.83 | 114.83 | 76.8 | 37.66 | 62.78 |
> | | SMART | 19.34 | 134.07 | 181.07 | 51.05 | 96.38 |
> | | MUR | 10.69 | 53.93 | 46.5 | 25.31 | **34.11** |
> | **LLM As Critic** | Per-Step Scale | 18.58 | 104.63 | 70.9 | 42.78 | 59.22 |
> | | Avg. Uncertainty | 15.65 | 64.27 | 71.17 | 40.34 | 47.86 |
> | | SMART | 15.94 | 49.47 | 61.63 | 21.98 | **37.26** |
> | | MUR | 12.84 | 63.43 | 56.4 | 20.98 | 38.41 |
> | **$\phi$-Decoding** | Per-Step Scale | 42.59 | 128.6 | 123.57 | 85.27 | 95.01 |
> | | Avg. Uncertainty | 19.70 | 109.3 | 74.53 | 36.23 | 59.94 |
> | | SMART | 28.75 | 149.07 | 137.2 | 52.68 | 91.93 |
> | | MUR | 20.83 | 94.87 | 92.73 | 23.79 | **58.06** |
>
> **Qwen3-8B**
>
> | Method | Variant | MATH | AIME24 | AIME25 | GPQA-D | Avg. |
> | :--- | :--- | :--- | :--- | :--- | :--- | :--- |
> | **CoT** | | 27.18 | 68.28 | 65.92 | 20.13 | 45.38 |
> | **Guided Search** | Per-Step Scale | 59.32 | 351.58 | 338.58 | 79.32 | 207.20 |
> | | Avg. Uncertainty | 43.28 | 188.9 | 179.73 | 47.31 | 114.81 |
> | | SMART | 55.64 | 214.95 | 207.56 | 70.48 | 137.16 |
> | | MUR | 46.68 | 173.93 | 172.6 | 60.86 | **113.52** |
> | **LLM As Critic** | Per-Step Scale | 36.41 | 123.18 | 108.81 | 47.08 | 78.87 |
> | | Avg. Uncertainty | 30.61 | 129.33 | 122.64 | 32.6 | 78.80 |
> | | SMART | 35.12 | 125.73 | 107.29 | 40.23 | 77.09 |
> | | MUR | 31.38 | 120.03 | 100.02 | 37.45 | **72.22** |
> | **$\phi$-Decoding** | Per-Step Scale | 87.90 | 397.23 | 390.91 | 84.37 | 240.10 |
> | | Avg. Uncertainty | 58.35 | 288.96 | 267.33 | 41.22 | 163.97 |
> | | SMART | 80.20 | 351.19 | 306.03 | 91.84 | 207.32 |
> | | MUR | 51.32 | 286.79 | 253.98 | 48.56 | **160.16** |
>
> > Concern 3-3: The authors provide very little description of their component designs
>
> Thanks for your suggestion. In the previous version, we have included some implementation details in the appendix. According to your suggestions, we reorganize them and add more details. New details about implementation, prompt, hyper-parameter, model selection and detector design are attached in **Appendix B** in new version.
> Notably, we also provide the code implementation for better reproducibility.

---

### Official Review · Reviewer_fxtu · 2025-11-01

**Soundness:** 2
**Presentation:** 3
**Contribution:** 2
**Rating:** 6
**Confidence:** 4

**Summary:**

This paper focuses on efficient and adaptive Test-Time Scaling (TTS) methods. The authors propose MUR (Momentum Uncertainty Guided Reasoning), which uses uncertainty quantification to determine whether each reasoning step requires further scaling. The uncertainty momentum acts as a moving average of stepwise uncertainty.

MUR is a training-free approach. The authors provide theoretical proofs demonstrating that MUR outperforms simple uncertainty-based methods by emphasizing recent steps, reducing variance, and achieving better convergence.

Extensive experiments show that MUR achieves superior results on mathematical reasoning tasks and GPQA. The authors also explore its performance across different models, TTS strategies, and threshold settings.

**Strengths:**

1. The proposed method improves both the efficiency and performance of test-time scaling without requiring any additional training.
2. The authors conducted extensive experiments across various TTS paradigms and multiple models to validate the effectiveness of their method.
3. The authors attempted theoretical derivations to demonstrate the effectiveness of MUR.

**Weaknesses:**

1. The theoretical analysis presented by the authors lacks some rigor. For example, the definition of noise in the uncertainty estimation is oversimplified — modeling it directly as standard Gaussian noise seems too simplistic. Moreover, the authors did not provide supporting references to justify this assumption.
2. The efficiency improvement achieved by the proposed MUR method, compared to the naive uncertainty averaging approach, is relatively limited in terms of the average number of tokens saved.
3. The paper lacks experiments on larger-scale models and a broader range of model families, such as Llama and others.

**Questions:**

In Section 3.2, the paper introduces the momentum uncertainty term M_t = α* M_{t-1} + (1-α)*m_t to represent the overall uncertainty along the reasoning trajectory. Since this formulation exponentially decays earlier uncertainty values, could it potentially diminish the influence of early but crucial reasoning steps that have low uncertainty but are logically important for later inference?

---

> ### Author Response · Authors · 2025-11-20
> **Response to Reviewer fxtu (Part 1/2)**
>
> Thanks for your support in extensive experiments, good performance and theoretical derivations.
>
> > Concern 2-1: The theoretical analysis presented by the authors lacks some rigor. The definition of noise in the uncertainty estimation is too simplistic.
>
> (1) We acknowledge that the Gaussian noise assumption is a simplification due to the inherent difficulty in precisely modeling the complex noise distribution in LLM generation. To enhance the rigor of this assumption, we supplemented our analysis with empirical evidence. We perform an autocorrelation function (ACF) experiment on real data generated from the Qwen3 series, showing that **we have confidence over 95% that the noise is not temporally correlated**. Based on this empirical finding, the assumption of Gaussian noise becomes a more justifiable and theoretically tractable approximation, which allows us to provide the core theoretical intuition behind the superiority of MUR. Furthermore, we include an extra experimental result to validate this theoretical intuition empirically. Details can be found in **Appendix A.3** in the new version.
>
> (2) To further clarify the theoretical superiority and give rigor proof, we model the role of momentum uncertainty as a low-pass filter[1], and provide more analysis on why this low-pass property works under uncertainty estimation setting. This property allows MUR to effectively remove high-frequency noise and sudden fluctuations in the uncertainty estimates, enabling us to better track the true evolution of reasoning uncertainty across steps. Details can be found in **Sec 3.2** and **Appendix A.2** in the new version.
>
> [1] On the Performance Analysis of Momentum Method: A Frequency Domain Perspective
>
> > Concern 2-2: About the efficiency of MUR compared with naive uncertainty averaging baseline.
>
> Averaged across all settings in Table 1, MUR saved about 8.09% tokens more than naive averaging approach. As a bonus, MUR achieves 1.66% accuracy gain than naive averaging approach. These empirical findings are significant across multiple benchmarks.
>
> What's more, MUR shows more stability than naive averaging approach, more robust to different model sizes and tasks. It is a big advantage in real applications.

---

> > ### Author Response · Authors · 2025-11-20
> > **Response to Reviewer fxtu (Part 2/2)**
> >
> > > Concern 2-3: Lack experiments on larger-scale models and a broader range of model families, such as Llama and others.
> >
> > According to your suggestion, we supplement more experiments on larger models and other model families: Qwen3-30B-A3B,  Glm4-9B. The results are presented as follows:
> >
> > **Results on Qwen3-30B-A3B**
> >
> > | Method | Variant | MATH | | AIME24 | | AIME25 | | GPQA | | Avg. | |
> > | :--- | :---| :--- | :--- | :--- | :---| :--- | :--- | :--- | :--- | :--- | :--- |
> > | **CoT** | | 83.40 | 606 | 26.67 | 3,104 | 20.00 | 3,182 | 43.94 | 579 | 43.50 | 1,868 |
> > | **Guided Search** | Per-Step Scale | 85.20 | 7,345 | 36.67 | 59,935 | 23.33 | 53,974 | 46.47 | 2,350 | 47.92 | 30,901 |
> > | | Avg. Uncertainty | 84.20 | 3,779 | 46.67 | 28,139 | 33.33 | 28,933 | 47.47 | 1,044 | 52.92 | 15,474 |
> > | | SMART | 85.40 | 5,601 | 40.00 | 39,829 | 30.00 | 39,232 | 45.96 | 1,879 | 50.34 | 21,635 |
> > | | MUR | 84.60 | 1,564 | 53.33 | 27,788 | 30.00 | 27,622 | 49.49 | 855 | **54.36** | **14,457** |
> > | **LLM As a Critic** | Per-Step Scale | 84.80 | 2,985 | 43.33 | 13,630 | 23.33 | 16,335 | 45.96 | 731 | 49.36 | 8,420 |
> > | | Avg. Uncertainty | 84.40 | 2,502 | 40.00 | 14,553 | 26.67 | 15,143 | 43.94 | 676 | 48.75 | 8,219 |
> > | | SMART | 83.40 | 1,057 | 43.33 | 16,118 | 30.00 | 14,661 | 47.98 | 626 | 51.18 | 8,116 |
> > | | MUR | 86.60 | 1,594 | 43.33 | 14,622 | 30.00 | 13,394 | 45.45 | 706 | **51.35** | **7,579** |
> > | **$\phi$-Decoding** | Per-Step Scale | 82.60 | 3,609 | 43.33 | 129,993 | 33.33 | 113,040 | 42.93 | 3,289 | 50.55 | 62,483 |
> > | | Avg. Uncertainty | 81.20 | 1,990 | 43.33 | 48,869 | 30.00 | 48,246 | 47.98 | 1,162 | 50.63 | **25,067** |
> > | | SMART | 82.80 | 3,052 | 40.00 | 91,896 | 33.33 | 82,686 | 41.41 | 2,163 | 49.39 | 44,949 |
> > | | MUR | 86.00 | 1,700 | 43.33 | 51,730 | 30.00 | 54,023 | 48.48 | 1,578 | **51.95** | 27,258 |
> >
> > **Results on GLM-4-9B**
> >
> > | Method | Variant | MATH | | AIME24 | | AIME25 | | GPQA | | Avg. | |
> > | :--- | :---| :--- | :--- | :--- | :---| :--- | :--- | :--- | :--- | :--- | :--- |
> > | **CoT** | | 45.20 | 473 | 6.67 | 714 | 3.33 | 820 | 28.79 | 608 | 21.00 | 654 |
> > | **Guided Search** | Per-Step Scale | 56.60 | 3,494 | 10.00 | 17,566 | 6.67 | 14,876 | 30.30 | 3,569 | 25.89 | 9,876 |
> > | | Avg. Uncertainty | 48.60 | 1,196 | 13.33 | 9,300 | 3.33 | 8,709 | 27.27 | 1,785 | 23.13 | **5,248** |
> > | | SMART | 53.40 | 2,304 | 20.00 | 14,395 | 3.33 | 15,706 | 31.82 | 3,029 | 27.14 | 8,859 |
> > | | MUR | 49.60 | 901 | 20.00 | 11,107 | 10.00 | 10,137 | 32.83 | 1,574 | **28.11** | 5,930 |
> > | **LLM As a Critic** | Per-Step Scale | 53.20 | 1,037 | 13.33 | 12,038 | 13.33 | 11,561 | 25.25 | 1,080 | 26.28 | 6,429 |
> > | | Avg. Uncertainty | 56.00 | 1,008 | 20.00 | 6,082 | 6.67 | 7,694 | 31.31 | 855 | 28.50 | **3,910** |
> > | | SMART | 48.60 | 579 | 20.00 | 8,993 | 10.00 | 10,998 | 25.75 | 711 | 26.09 | 5,320 |
> > | | MUR | 53.20 | 746 | 23.33 | 8,786 | 13.33 | 10,373 | 29.80 | 890 | **29.92** | 5,199 |
> > | **$\phi$-Decoding** | Per-Step Scale | 44.60 | 3,817 | 16.67 | 26,268 | 13.33 | 56,546 | 25.25 | 3,495 | 24.96 | 22,532 |
> > | | Avg. Uncertainty | 47.80 | 1,316 | 23.33 | 21,796 | 6.67 | 19,333 | 29.80 | 1,466 | 26.90 | **10,978** |
> > | | SMART | 47.00 | 2,386 | 20.00 | 41,494 | 16.67 | 47,241 | 28.79 | 3,791 | 28.12 | 23,728 |
> > | | MUR | 48.00 | 1,531 | 23.33 | 26,966 | 10.00 | 24,520 | 31.31 | 2,466 | **28.16** | 13,871 |
> >
> > > Concern 2-4: Since this formulation exponentially decays earlier uncertainty values, could it potentially diminish the influence of early but crucial reasoning steps that have low uncertainty but are logically important for later inference?
> >
> > Thanks for raising the question.
> >
> > (1) We clarify that the core of MUR is to model the reasoning uncertainty and scale computes dynamically, rather than focus on the logical information.
> >
> > (2) Momentum uncertainty is introduced to measure the overall uncertainty of reasoning progress, instead of the specific steps. In this context, the most recent steps matter than the early steps, because recent steps can track the overall uncertainty "changes" more precisely. The scaling is triggered based on the changes (i.e., comparison between previous momentum uncertainty and current step uncertainty).

---

### Official Review · Reviewer_ybr6 · 2025-11-02

**Soundness:** 3
**Presentation:** 2
**Contribution:** 2
**Rating:** 4
**Confidence:** 3

**Summary:**

This paper addresses the "overthinking" problem in Test-Time Scaling (TTS). It proposes MUR (Momentum Uncertainty-guided Reasoning), a method inspired by the concept of "momentum" in physics, which dynamically allocates the computational budget to critical reasoning steps. MUR is claimed to reduce computation by an average of about 45% across multiple benchmarks while also yielding an accuracy improvement of 0.33% to 3.46%.

**Strengths:**

- The problem is well-defined. The "Per-Step Scale" baseline is demonstrably inefficient, and the pursuit of a training-free, adaptive strategy to guide TTS is an excellent and practical research objective.

- The method acts as a universal "wrapper," demonstrating compatibility and benefits when combined with various existing TTS techniques (including Guided Search, LLM as Critic, $\phi$-Decoding, and Thinking Mode). This broad applicability is a significant practical advantage.

**Weaknesses:**

- The reported accuracy increase ($0.33\% \text{ to } 3.46\%$) is suspicious. It is unclear if these "improvements" are simply experimental noise. The paper never reports the standard deviation from multiple runs. In LLM reasoning tasks, a $0.3\%$ improvement is easily within the range of expected stochastic variability.

- The paper claims to provide "deep theoretical proof" (Section 3.2 and Appendix A). However, these "proofs" merely reiterate the well-known properties of Exponential Moving Average (EMA), specifically that it reduces variance and bias compared to the raw signal. This is a fundamental concept in signal processing and statistics, not an innovative theoretical contribution to the field of reasoning. Furthermore, this proof only establishes EMA as a stable estimator, not that MUR provably leads to better reasoning outcomes.

**Questions:**

- The "accuracy improvement" is the paper's core and most surprising claim. Given the inherent randomness of LLM inference, are these gains statistically significant? Could the authors please provide the standard deviation from multiple runs for the key accuracy results presented in Tables 1 and 2?

- The paper argues that MUR avoids "overthinking." Assuming that a "Per-Step Scale" baseline is correctly implemented, why would this "overthinking" damage accuracy (rather than just waste computation)? It seems counter-intuitive that reducing work would lead to better results. Is it possible that the gain is merely the result of avoiding the accuracy degradation caused by excessive scaling?

---

> ### Author Response · Authors · 2025-11-20
> **Response to Reviewer ybr6 (Part 1/2)**
>
> Thanks for your thoughtful and positive assessment of our work on the well-defined research problem, generalizable methods and broad applicability.
>
> > Concern 1-1: About the performance improvement and standard deviation from multiple runs.
>
> Firstly, the 0.33\% performance gain you cited is merely the minimal result observed from one specific experimental setup (Qwen3-4B + Guided-Search). When averaging the results across all experimental configurations, the mean improvement is 1.66\%, which constitutes a significant overall gain across multiple benchmarks.
>
> Secondly, it is crucial to note that the reported improvements are benchmarked against the *Per-Step-Scale* baseline, which is typically regarded as the performance ceiling for this task. More importantly, while we observe performance gains, they are not the primary claim of this work. Our main objective is the pursuit of more efficient compute allocation while maintaining performances; therefore, the extra observed gains are an extra benefit (a bonus).
>
> Thirdly, according to your suggestions, we conduct five runs for the key accuracy of MUR in Table 1 and 2, and report the result with standard deviation:
>
> | Model Size | Method | Metric | MATH-500 | AIME24 | AIME25 | GPQA-D | Avg. |
> | :---: | :---: | :---: | :---: | :---: | :---: | :---: | :---: |
> | **1.7B** | **Guided Search** | std | 0.53 | 3.37 | 0.73 | 2.81 | 0.42 |
> | | | mean | 71.40 | 20.28 | 9.93 | 29.80 | 32.85 |
> | | **LLM as Critic** | std | 0.53 | 1.80 | 1.18 | 1.54 | 0.34 |
> | | | mean | 70.60 | 20.42 | 11.18 | 30.64 | 33.21 |
> | | **$\phi$-Decoding** | std | 0.72 | 0.12 | 1.69 | 0.58 | 0.43 |
> | | | mean | 70.60 | 20.14 | 10.55 | 26.93 | 32.06 |
> | | **Thinking Switch** | std | 1.40 | 3.28 | 1.20 | 0.58 | 0.59 |
> | | | mean | 87.73 | 50.63 | 32.64 | 39.56 | 52.64 |
> | **4B** | **Guided Search** | std | 0.42 | 0.72 | 1.92 | 0.29 | 0.58 |
> | | | mean | 81.87 | 29.16 | 20.28 | 41.75 | 43.27 |
> | | **LLM as Critic** | std | 0.42 | 2.46 | 1.09 | 1.17 | 0.71 |
> | | | mean | 82.07 | 28.12 | 18.96 | 40.24 | 42.35 |
> | | **$\phi$-Decoding** | std | 0.20 | 1.37 | 0.12 | 0.51 | 0.23 |
> | | | mean | 79.40 | 28.54 | 18.26 | 41.41 | 41.90 |
> | | **Thinking Switch** | std | 1.06 | 0.48 | 0.72 | 3.21 | 0.86 |
> | | | mean | 92.80 | 67.85 | 59.38 | 55.89 | 68.98 |
> | **8B** | **Guided Search** | std | 0.58 | 2.92 | 0.36 | 1.54 | 0.94 |
> | | | mean | 83.53 | 37.99 | 24.59 | 45.29 | 47.86 |
> | | **LLM as Critic** | std | 0.35 | 1.73 | 0.48 | 2.78 | 0.93 |
> | | | mean | 84.20 | 33.61 | 22.22 | 46.80 | 46.71 |
> | | **$\phi$-Decoding** | std | 0.23 | 1.44 | 0.84 | 1.75 | 0.45 |
> | | | mean | 84.53 | 35.84 | 24.24 | 46.46 | 47.77 |
> | | **Thinking Switch** | std | 0.00 | 1.16 | 1.03 | 1.82 | 0.31 |
> | | | mean | 93.80 | 72.08 | 61.74 | 58.08 | 71.42 |
>
>
> > Concern 1-2: About the contribution of the theoretical part.
>
> We agree with the reviewer that our proposed momentum uncertainty shares a structural similarity with the Exponential Moving Average (EMA). However, we respectfully highlight the novelty and contribution of its application as follows:
>
> 1. **Non-Trivial Application**: While EMA is a fundamental concept applied across various domains, the exploration and validation of its properties within the complex and specific dynamics of LLM reasoning is non-trivial. Our method, MUR, is thoroughly motivated by the successful application of Gradient Descent with Momentum. We argue that our contribution, as the first work to successfully apply and model this concept to stabilize and improve performance in the LLM reasoning setting, represents a significant advancement.
>
> 2. **Theoretical Clarity & Novelty**: To further clarify the technical novelty and contribution beyond the EMA formula, we have significantly supplemented the explanations. Specifically, we illustrate the critical role of momentum uncertainty as a *low-pass filter* under the reasoning settings. Please refer to the updated paper for details (for example, lines 207-213, and 852-857).
>
> 3. **Theoretical Support for Better Performance**: To provide stronger theoretical support for the claim that MUR leads to better reasoning outcomes, we have included **two new experiments**. These studies combine real data analysis with our theoretical intuition, providing concrete evidence for MUR's effectiveness in reasoning tasks. The results are detailed in **Appendix A.3** of the new version.

---

> > ### Author Response · Authors · 2025-11-20
> > **Response to Reviewer ybr6 (Part 2/2)**
> >
> > > Concern 1-3: Why would this "overthinking" damage accuracy ？
> >
> > Thanks for your insightful question. It precisely captures our key motivation.
> >
> > We fully agree that the observed performance gain primarily stems from mitigating the accuracy degradation caused by indiscriminate or excessive scaling, a phenomenon also noted in recent literature (e.g., [2, 3]).
> >
> > Also, we provide a simple intuition here: (1) *Scaling Simple Steps.* Optimizing simple (low-uncertainty) reasoning steps is often **non-beneficial** because the LLM usually generates them correctly initially. Crucially, **excessive scaling introduces the risk of flipping a correct step into an incorrect one (i.e., accuracy degradation)**. (2) *Scaling Difficult Steps.* Optimizing difficult (high-uncertainty) steps is generally harmless and beneficial, as the LLM often generates them incorrectly initially. Scaling has the potential to **correct an incorrect step**.
> >
> > Since a higher uncertainty inherently indicates that the LLM is struggling to generate the correct answer for that step [4, 5], our Momentum Uncertainty Reasoning (MUR) strategy only applies scaling to those steps where the uncertainty level suggests difficulty. This targeted approach effectively avoids performance degradation linked to excessive scaling.
> >
> > ### Reference
> >
> > [1] On the Performance Analysis of Momentum Method: A Frequency Domain Perspective
> >
> > [2] Dynamic Early Exit in Reasoning Models
> >
> > [3] Sampling-efficient test-time scaling: Self-estimating the best-of-n sampling in early decoding
> >
> > [4] Response Uncertainty and Probe Modeling: Two Sides of the Same Coin in LLM Interpretability?
> >
> > [5] Uncertainty Quantification and Decomposition for LLM-based Recommendation

---

### Author Response · Authors · 2025-11-27

Dear Reviewers,

Thanks for your time in reviewing our paper. Previously, we have supplemented the response to your concerns, and also uploaded the updated version. Since the discussion phase is about to close, would you please take a moment to check whether our responses have adequently addressed your concerns? Thanks for your time and effort.

Best,

Authors of MUR

---

### Note · Authors · 2025-12-30

**Comment:**

Dear AC, reviewers,

Thank you all for your hard work and careful review. We are really grateful of your inspiring suggestions and  polish our paper based on them.  Additionally, we try to address your questions and concerns with detailed explanations and experiments,  but you cannot reply due to the sudden stop of the rebuttal phase. Considering this, we decide to withdraw our paper.

Best wishes,
Authors

**Withdrawal Confirmation:**

I have read and agree with the venue's withdrawal policy on behalf of myself and my co-authors.